# FROM CATEGORIES TO CLASSIFIER: NAME-ONLY CONTINUAL LEARNING BY EXPLORING THE WEB

## ABSTRACT

Continual Learning (CL) often relies on the availability of extensive annotated datasets, an assumption that is unrealistically time-consuming and costly in practice. We explore a novel paradigm termed *name-only continual learning* where time and cost constraints prohibit manual annotation. In this scenario, learners adapt to new category shifts using only category names without the luxury of annotated training data. Our proposed solution leverages the expansive and ever-evolving internet to query and download *uncurated* webly-supervised data for image classification. We investigate the reliability of our web data and find them comparable, and in some cases superior, to manually annotated datasets. Additionally, we show that by harnessing the web, we can create support sets that surpass state-of-the-art name-only classification that create support sets using generative models or image retrieval from LAION-5B, achieving up to 25% boost in accuracy. When applied across varied continual learning contexts, our method consistently exhibits a small performance gap in comparison to models trained on manually annotated datasets. We present *EvoTrends*, a class-incremental dataset made from the web to capture real-world trends, created in just minutes. Overall, this paper underscores the potential of using uncurated webly-supervised data to mitigate the challenges associated with manual data labeling in continual learning.

## 1 INTRODUCTION

Continual Learning (CL) predominantly rely on annotated data streams, i.e., a common underlying assumption is the availability of well-curated, annotated datasets. However, the financial and temporal costs associated with continual annotation is staggering. To illustrate this, annotating 30K samples in the CLEAR10 dataset (Lin et al., 2021), a popular CL dataset, despite using optimized annotation workflows with large CLIP models (Radford et al., 2021), cost $4,500 and more than a day worth of annotation time. In contrast, businesses like Amazon and Fast Fashion companies constantly need to update their image classification models and associated recommendation engines due to changing inventory, seasonal and customer trends. Annotating labeled training sets every time for commercial classification models with thousands of categories and millions of samples is unrealistic, as it would take weeks and cost hundreds of thousands of dollars. In short, manual data collection and annotation are expensive and time-consuming, posing a bottleneck in real-world continual learning.

To this end, we explore a new scenario called *name-only continual learning*[1]. As commonly done in the traditional continual learning, new categories or domain shifts are continuously introduced at each timestep and we need to quickly adapt the classification model to the changes in the stream; however, in this setting we cannot create annotated training datasets. At each timestep, the learner is only provided with category/class names and is allocated a computational budget to adapt to the new classes. At the end of each timestep, the learner is presented with test samples and its performance is assessed. To tackle this setting, we propose to leverage the ever-evolving internet by query and downloading uncurated webly-supervised data for continual image classification. This will dramatically speed up the process of continually updating classifiers, from once in several days to once practically every hour.

*Why Revisit Webly-Supervised Learning* (Fergus et al., 2005; Schroff et al., 2010)? Recently, countries like Japan[2] have enacted legislations allowing the use of online data for training deep models,

---

[1]We borrow the term name-only classification from (Udandarao et al., 2023). We do not use zero-shot classification (Lampert et al., 2009) as it aims to generalize to unseen categories *without seeing any examples*, using attribute information whereas name-only setting allows access to public models and data.

[2]https://aibusiness.com/data/japan-s-copyright-laws-do-not-protect-works-used-to-train-ai-

irrespective of the copyright status. This follows the intuition that one can learn and be inspired from copyrighted materials so long they do not regenerate it or derivate works, such as with classification models. This allows us to leverage the internet, which functions as an ever-expanding database, continually updating itself with billions of new photos daily, staying current with the latest trends. Additionally, it provides search engines that traditionally offer highly relevant image results at scale, allowing us to query and download webly-supervised data cheaply and in just minutes. Being dynamic, the internet is ideal for continually updating to rapid changes in the stream. In this context, we address three crucial questions about the use of the web for training dataset creation:

❶ *How reliable is our uncurated webly-supervised data?* To assess its quality, we compare performance of deep learning models on our webly-supervised training data with manually annotated datasets for fine-grained image classification, which typically require expert annotations. We find that in some cases models trained on uncurated webly-supervised data can equal or even surpass the performance of those trained on manually annotated datasets. We show that this performance primarily results from our ability to cheaply gather much larger training sets than manual annotation allows.

❷ *How does uncurated webly-supervised data compare to the latest name-only classification approaches?* We demonstrate that using uncurated webly-supervised data, one can outperform alternative methods of dataset generation used in state-of-the-art name-only classification approaches (Udandarao et al., 2023; He et al., 2022; Wallingford et al., 2023) on the same CLIP model by an impressive 5-25% absolute accuracy improvement. Our approach can also generalize to vision-only self-supervised models like MoCoV3 ImageNet1K models (Chen et al., 2021).

❸ *Can we efficiently utilize uncurated webly-supervised data across various continual learning settings?* We apply our name-only webly-supervised approach to various continual learning situations such as class-incremental (new classes introduced over time), domain incremental (new domains introduced over time), and time incremental (mimicking a chronologically ordered class-annotated stream). In each of the above scenarios where we had access only to class names, our models trained on uncurated webly-supervised data only had a small performance gap compared to those trained on curated datasets. To illustrate our capabilities beyond existing datasets, we introduce *EvoTrends*, a continual learning dataset that introduces trending products year-by-year from 2000 to 2020. This underscores our ability to build classifiers and deploy them in a continual manner within minutes without relying on manually curated training datasets.

In summary, our primary contributions address the aforementioned three questions, conclusively showing that using uncurated webly-supervised data can significantly reduce the time and expense associated with manual annotation in the proposed name-only continual learning setting.

## 2 NAME-ONLY CONTINUAL LEARNING: PROBLEM FORMULATION

In the *name-only* classification setup, the target is to learn a function $f_\theta$ parameterized by $\theta$, where here, unlike traditional classification tasks, the only given information is the class categories denoted by $\mathcal{Y}$. While additional context about the data distribution (*e.g.* cartoon, art, sketch,...) is allowed to be given in $\mathcal{Y}$, no training samples are provided. In contrast to the zero-shot setting, the learner is allowed to use publicly available data and models, with the exception of the original training set and models trained on it. For example, the use of prominent backbones like GPT (OpenAI, 2023), DALL-E (Ramesh et al., 2022) and assembling a training set from public datasets such as LAION5B (Schuhmann et al., 2022) is allowed to obtain the classifier. The performance of the learner is subsequently assessed on a curated test set, $\mathcal{X}^*$.

We extend the *name-only* classification paradigm to continual learning, dubbing this *name-only continual learning*. In this setup, we perform *name-only* classification across multiple timesteps, $t \in \{1, 2, 3, \dots\}$. For each timestep $t$, a data stream $\mathcal{S}$, unveils a distinct set of class categories, $\mathcal{Y}_t$. Notably, $\mathcal{Y}_t$ might introduce categories absent in preceding timesteps; that is, a category $y_t \in \mathcal{Y}_t$ might not belong to $\mathcal{Y}_j$ for all $j < t$. Subsequently, at each $t$, the algorithm must continually update the classifier $f_\theta$ by using prominent backbones or publicly available data.

Formally, the primary goal in continual learning, is to learn a classifier $f_{\theta_t} : \mathcal{X} \to \bigcup_{i=1}^{t} \mathcal{Y}_i$, parameterized by $\theta_t$, that correctly classifies a category from all the introduced class categories up to the current timestep. Given that evaluation samples could originate from any past class categories, *i.e.* $y_i \in \bigcup_{i=1}^{t} \mathcal{Y}_i$, the updated model $f_{\theta_t}$ must maintain its capabilities in classifying earlier seen classes. In summary, at every timestep $t$:

1. The data stream, $\mathcal{S}$, presents a set of categories, $\mathcal{Y}_t$, to be learned.

2. Under a given computational budget, $\mathcal{C}_t$, the classifier $f_{\theta_{t-1}}$ is updated to $f_{\theta_t}$.

3. To evaluate the learner, the stream $\mathcal{S}$ presents test samples $\{(\mathbf{x}_i, y_i)\}_{i=1}^n$ with $y_i$ belonging to the collective set $\bigcup_{i=1}^t \mathcal{Y}_i$.

In Step 3, it is important to note that the annotated test set is reserved solely for evaluation. Neither the images nor the labels from the test set are available for the model in any future training steps. Moreover, it is worth noting that computational budgeting has become the prevailing standard in CL Prabhu et al. (2023a). This practice involves setting limits, either in terms of computation or time, hence on the number of samples that can be generated or annotated for training purposes.

## 3   OUR APPROACH: CATEGORIES TO CLASSIFIER BY EXPLORING THE WEB

Without access to training data, one might be tempted to use generative models to create training data. However, as explained in Section 2, the continual learner is constrained by a budget limit $\mathcal{C}_t$. This budget constraint makes generative methods computationally impractical due to their high computational requirements. Hence, we discuss our approach, *"C2C"*, for transitioning from class categories to classifiers within a computational budget. At each timestep $t$, our approach involves takes main steps: (1) collecting data from the web, which we refer to as uncurated webly-supervised data and (2) training a classifier using this data.

**Step 1.  Querying and Downloading Uncurated Webly-Supervised Training Data.**  There are several challenges associated with querying the web which raises questions that we address below:

*How to design web queries?*  The web is expansive and noisy, and simply class categories provided by stream are often not specific enough. Consider the category name "snapdragon": on its own, search engines might yield images of computer chips. Hence, we design a simple querying strategy of adding an auxiliary suffix to refine our queries. Our searches follow the pattern: `Category Name + Auxiliary Suffix`. When building a flower dataset and querying "snapdragon", appending the suffix "flower" refines the query to focus on the desired botanical images. Moreover, within domain-incremental settings, we can adapt our search by using domain-specific suffixes like "cartoon" for cartoon images. In summary, this addition offers a richer context, steering the search engine more precisely.

*How do we prevent unintentional download of explicit images?*  Past webly supervised methods have unintentionally collected explicit content from online sources (Birhane & Prabhu, 2021). To address this, we implemented some cost-effective safeguards. First, we enabled strict safe-search feature on our search engines, which helps filter out explicit or inappropriate content. Second, we ensure that class-categories $\mathcal{Y}_t$ do not have explicit terms by manually checking the queries and replacing possible offensive terms with less offensive ones, e.g. "african ass" would be replaced by "african wild donkey" or "Equus africanus". We manually inspected a few hundred of the downloaded images with random sampling and found no explicit content providing preliminary evidence of effectiveness of the safeguards.

*Improvements in the speed of querying and download.*  The end-to-end scraping and downloading time required for 39 million flickr samples in a stress test required 12 days using a standard Python query and download pipeline. We optimized and reduced it to just 2 days - a 600% improvement - using the same computational resources. We applied the same pipeline to accelerate querying and downloading of uncurated internet data, we utilize parallelization across multiple dimensions: (1) We query four major search engines concurrently - Bing, Flickr, Google and DuckDuckGo - using separate CPU nodes in a cluster. This allows for simultaneous querying across engines. (2) We use an efficient multi-threaded querying tool[3] that handles image search queries in parallel for each engine. This tool utilizes FIFO threaded queues to concurrently manage the search and download workflows for each query. (3) After aggregating image links from different engines, we leverage a parallelized image downloading tool[4], which additionally applies postprocessing such as resizing. In conclusion, the key factors were concurrent querying across multiple search engines, fast multi-threaded querying per engine, and parallelized downloading and resizing of images.

---

[3] https://github.com/hellock/icrawler
[4] https://github.com/rom1504/img2dataset

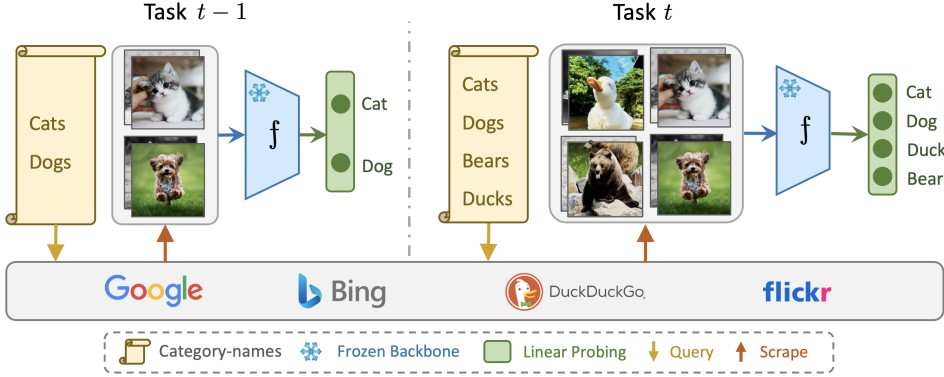

Figure 1: **Continual Name-Only Classification: Our Approach.** At each timestep $t$, the learner receives a list of class categories without any training samples. We start by collecting webly-supervised data through querying and downloading data from multiple search engines. We then extract features using a frozen backbone, and subsequently train a linear layer on those features. The same process is repeated for the next timestep.

**Step 2. Classifier Training.** Once we have uncurated webly-supervised data the next step is to train a classifier. At each timestep $t$, the learner is assigned a computational budget, denoted as $\mathcal{C}_t$. Ideally, this budget should include the entire data collection process, whether it involves querying and downloading from the web or manual annotation. It is important to note that including this overhead within the budget would make it challenging or even impossible for manually annotated datasets to receive sufficient training, as their annotation pipeline incurs significant costs. We test three budgets: tight, normal, and relaxed. Normal budget allows for a training equivalent to 1 epoch on the first timestep of the manually annotated datasets (details in Appendix D). The "tight" budget is half of the normal, while the "relaxed" budget is four times the normal, as done in (Prabhu et al., 2023a). Under this budgeted setup, we compare three continual learning baselines (a) Linear Probing, (b) NCM (Mensink et al., 2013; Janson et al., 2022) and (c) KNN (Malkov & Yashunin, 2018; Prabhu et al., 2023b), providing insights into efficient CL methods with fixed feature extractor.

Our approach is summarized in Figure 1. In our *continual name-only* classification setting, for each timestep $t$, we query and download webly-supervised data based on the provided class categories $\mathcal{Y}_t$, by following the recipe described in Step 1. Once we complete downloading the data, the classifier is trained not only on the uncurated data downloaded from the current timestep $t$ but also on the uncurated data downloaded from all prior timesteps.

## 4    EVALUATING CAPABILITIES OF UNCURATED WEBLY-SUPERVISED DATA

We begin by examining two main questions: ❶ How reliable is uncurated webly-supervised data? Specifically, can models trained on these training sets match the accuracy of those trained on expert-annotated training sets? ❷ How does the uncurated webly-supervised data compare to the latest name-only classification approaches? For instance, can models trained on our data surpass the latest methods tailored for vision-language models, such as CLIP, in a name-only classification context? We analyze these questions in the better studied non-continual name-only classification setting where one is provided with a set of class categories to be learnt.

### 4.1    EXPERIMENTAL DETAILS

**Datasets.** We focus primarily on fine-grained classification tasks due to two main reasons: (i) such tasks present a greater challenge than coarse-grained datasets especially when sourced from noisy data sources such as the web, and (ii) they are prevalently employed in name-only classification benchmarks, facilitating comprehensive comparisons with existing methods. We evaluate the classification accuracy across five benchmarks that contain a broad selection of classes: (1) FGVC Aircraft (Maji et al., 2013), (2) Flowers102 (Nilsback & Zisserman, 2008), (3) OxfordIIITPets (Parkhi et al., 2012), (4) Stanford Cars (Krause et al., 2013), and (5) BirdSnap (Berg et al., 2014).

**Models.** We use a fixed backbone, ResNet50 MoCoV3 (Chen et al., 2021), and experiment with two classifiers on top: (i) Linear Probe and (ii) MLP Adapter. The MLP Adapter consists of a three-layer model: `input_dim` $\rightarrow 512$, $512 \rightarrow 256$, and $256 \rightarrow$ `num_classes`, with Dropout(0.5) and ReLU nonlinearities. Additionally, we also try fine-tuning both the backbone and a linear layer.

Table 1: **Performance Analysis between Uncurated Webly-Supervised Data (C2C) and Manually Annotated Training (MA) Data.** Despite utilizing uncurated web data, our results demonstrate competitive or even better performance than that of manually annotated datasets in fine-grained categorization tasks. The most notable improvements are observed when using MLP-adapters.

| | | Datasets | | | | |
|---|---|---|---|---|---|---|
| Evaluation | Training Dataset | FGVC Aircraft | Flowers102 | OxfordIIITPets | Stanford Cars | BirdSnap |
| Linear | MA Data | 38.5% | 83.3% | 89.8% | 56.3% | 46.2% |
| Probe | C2C (Ours) | 57.5% (+19.0%) | 85.7% (+2.4%) | 91.7% (+1.9%) | 62.1% (+5.8%) | 56.1% (+9.9%) |
| MLP | MA Data | 46.0% | 80.3% | 89.7% | 57.6% | 47.7% |
| Adapter | C2C (Ours) | 65.5% (+19.5%) | 87.1% (+6.8%) | 92.8% (+3.1%) | 66.8% (+9.2%) | 53.7% (+6.0%) |
| Finetune | MA Data | 76.6% | 94.3% | 92.8% | 91.6% | 70.4% |
| Backbone | C2C (Ours) | 94.8% (+18.2%) | 93.3% (-1.0%) | 94.7% (+1.9%) | 92.8% (+1.2%) | 69.9% (-0.5%) |

**Training Procedure.** For linear probing and MLP adapter experiments, we freeze the backbone and extract features from both our uncurated webly-supervised data and manually annotated (MA) datasets. We then perform linear probing and MLP adapter training on the extracted features. Our training uses an Adam optimizer with a batch size of 512 and a learning rate of 0.001. We use a LRonPlateau scheduler with a patience of 10 and a decay of 0.1. Models are trained for 300 epochs, reaching convergence within 10-50 epochs. For finetuning experiments for both our uncurated webly-supervised data and manually annotated (MA) datasets, we use an SGD optimizer with a learning rate of 0.1 and a linear learning rate scheduler. A batch size of 128 and standard data augmentations are applied. Models are trained until convergence on both uncurated web data and the manually annotated training sets, within 50 epochs for our uncurated web data and up to 250 for manually annotated datasets. Class-balanced random sampling is used for all experiments, especially helpful for data downloaded from the internet given its natural long-tail distribution.

## 4.2 How Reliable Is Uncurated Webly-Supervised Data?

We begin by addressing our first fundamental question: ❶ Can uncurated webly-supervised data serve as a substitute for meticulously curated training data? Put simply, can web data match the performance of manually annotated datasets?

**Results and Key Findings.** Table 1 contrasts the performance of our uncurated webly-supervised data with manually annotated datasets. Remarkably, classifiers trained on our webly-supervised data surpass those trained on manually annotated datasets by a margin of $1 - 19\%$ (highlighted in green). In the worst-case scenario, there is a performance decline of less than $1\%$ (marked in red). The most pronounced improvement, ranging from $3 - 19\%$, arises when our webly-supervised data is integrated with an MLP-Adapter. As anticipated, fine-tuning yields superior results compared to the MLP adapter, which in itself outperforms linear probing.

In summary, using uncurated webly-supervised data consistently outperform manually annotated datasets across different classifiers and datasets. This finding is counterintuitive, given that our web data is: (i) uncurated, (ii) noisy, and (iii) out-of-distribution with respect to the test set. The reason behind this apparent paradox can be attributed to the dataset sizes. Details are provided in the Appendix B and summarized below.

**How Does Scale Influence Our Performance?** Our webly-supervised datasets are notably large due to cheap query and download properties, being approximately 15 to 50 times larger than the manually annotated datasets. Hence, we explore the impact of scale by limiting our queries to search engines to return only the top-$k$ images in Table 2. Our results suggest that query size is the primary driver for performance gains. When we limit our query size to match the size of manually annotated datasets (using the top 10 or 20 images per engine per class), there is a drop in accuracy by 10-20% relative to manually curated datasets. However, as we gather more data, we consistently observe performance improvements. The scalability of our method, only possible due to virtually no scaling cost. The primary cause of superior performance is by scaling the size of downloaded data, without high costs of manual annotations or other checks.

In Appendix B, we explore various factors that, surprisingly, had little to no impact on the effectiveness of our approach. Our approach demonstrated strong performance across various model architectures and training protocols. Its strength was mostly evident when sourcing data from multiple web engines (Google, Flickr, DuckDuckGo Bing), effectively handling diverse data distributions. Surprisingly, even after cleaning our web data using deduplication and automatic removal of noisy samples, reducing the data size by 30%, the performance remained unaffected. This suggests that the main challenges are likely due to out-of-domain instances rather than reducing noise or duplicate samples. Lastly, class-balanced sampling does not affect the performance of our

Table 2: **Impact of Size of Queried Webly-Supervised Data on Performance.** This table illustrates the influence of downsizing our queried web data by considering only the top-$k$ queries for download. Notably, a substantial performance drop occurs as the dataset size decreases. Despite the higher quality of the top-$k$ samples, their limited quantity adversely affects performance. We use Manually Annotated Training (MA) Data as a reference point.

| Eval. | Dataset Size | Datasets | | | | |
|---|---|---|---|---|---|---|
| | | FGVC Aircraft | Flowers102 | OxfordIIITPets | Stanford Cars | BirdSnap |
| Linear Probe | MA Data | 38.5% | 83.3% | 89.8% | 56.3% | 46.2% |
| | Top-10/engine/class | 18.3% ( -20.2%) | 64.3% ( -19.0%) | 82.0% ( -7.8%) | 34.3% ( -22.0%) | 36.4% ( -9.8%) |
| | Top-20/engine/class | 33.8% ( -4.7%) | 71.7% ( -11.6%) | 87.3% ( -2.5%) | 45.7% ( -10.6%) | 42.1% ( -4.1%) |
| | Top-50/engine/class | 40.8% (+2.3%) | 77.7% ( -5.6%) | 88.7% ( -1.1%) | 57.9% (+1.6%) | 48.5% (+2.3%) |
| | Top-100/engine/class | 52.4% (+13.9%) | 80.8% ( -2.5%) | 90.1% (+0.3%) | 64.6% (+8.3%) | 52.4% (+6.2%) |
| | Top-200/engine/class | 56.6% (+18.1%) | 82.7% ( -0.6%) | 90.7% (+0.9%) | 67.8% (+11.5%) | 54.6% (+8.4%) |
| | All Data | 57.5% (+19.0%) | 85.7% (+2.4%) | 91.7% (+1.9%) | 62.1% (+5.8%) | 56.1% (+9.9%) |
| MLP-Adapter | MA Data | 46.0% | 80.3% | 89.7% | 57.6% | 47.7% |
| | Top-10/engine/class | 32.2% ( -13.8%) | 66.0% ( -14.3%) | 86.7% ( -3.0%) | 39.8% ( -17.8%) | 36.4% ( -11.3%) |
| | Top-20/engine/class | 37.4% ( -8.6%) | 72.6% ( -7.7%) | 88.1% ( -1.6%) | 49.5% ( -8.1%) | 42.3% ( -5.4%) |
| | Top-50/engine/class | 51.4% (+5.4%) | 77.3% ( -3.0%) | 89.9% (+0.2%) | 61.1% (+3.5%) | 47.9% (+0.2%) |
| | Top-100/engine/class | 58.2% (+12.2%) | 81.9% (+1.6%) | 90.6% (+0.9%) | 65.8% (+8.2%) | 50.1% (+2.4%) |
| | Top-200/engine/class | 63.4% (+17.4%) | 84.1% (+3.8%) | 91.9% (+2.2%) | 70.3% (+12.7%) | 54.4% (+6.7%) |
| | All Data | 65.5% (+19.5%) | 87.1% (+6.8%) | 92.8% (+3.1%) | 66.8% (+9.2%) | 53.7% (+6.0%) |

model, indicating that further exploration of long-tailed loss functions (Karthik et al., 2021), may not yield significant improvements.

## 4.3 Comparison with Name-Only Classification Strategies

We now address our second question: ❷ How does the performance of webly-supervised datasets compare to the latest name-only classification approaches? Can web data surpass the latest methods tailored for vision-language models, such as CLIP, in a name-only classification context?

**Comparison with Recent Advances.** Traditional name-only classification methods are often built upon zero-shot CLIP (CLIP-ZS) (Radford et al., 2021). CLIP-ZS works by using text prompts that contain category names to classify images. For each test data point, it predicts the class by finding the category name prompt that best matches the input image. Recent research has introduced improvements to this approach in three main areas:

*(i) Better Text Prompts:* Methods like VisDesc (Menon & Vondrick, 2022), CuPL (Pratt et al., 2023) and WaffleCLIP (Roth et al., 2023) have explored more effective text prompts to enhance classification accuracy; *(ii) Creating Pseudo-training Datasets:* Approaches such as Glide-Syn (He et al., 2022) and Sus-X (Udandarao et al., 2023), and Neural Priming (Wallingford et al., 2023) focus on creating training datasets either by retrieval from LAION5B or generating samples from diffusion models to improve model performance, with retrieval being better (Burg et al., 2023); *(iii) Enhanced Adapters:* CALIP (Guo et al., 2023) , along with Glide-Syn (He et al., 2022) and Sus-X (Udandarao et al., 2023) propose improved adapters for CLIP models to enhance their classification abilities. There are alternative approaches, like SD-Clf (Li et al., 2023a), which shows the effectiveness of stable-diffusion models for classification tasks. Additionally, CaFo (Zhang et al., 2023) explores chaining different foundation models for tasks including name-only classification. We describe these approaches in detail in Appendix C.

**Results.** To evaluate our approach, we compare it against recent strategies using the ResNet50 CLIP model for a fair comparison. The results are summarized in Table 3; comparisons on CLIP ViT-B/16 model can be found in Appendix C. Consistently, our approach outperforms other leading methods such as CaFo and SuS-X-LC, with performance improvements between 2-25%. Additionally, we apply our apporach to vision-only ResNet50 MoCoV3 model trained on ImageNet1K. Notably, this often performs significantly better out-of-the-box than CLIP variants, with impressive improvements of 2-8%, offering new insights on recent works (Li et al., 2023b). Moreover, employing an MLP Adapter results in a 1-4% boost in performance over linear probing, and this is achieved with minimal added computational cost when compared to extracting features from a ResNet50 model.

*Why Does Our Webly-Supervised Data Outperform Other Approaches?* A fundamental factor in the superior performance of our approach is again the scale of our uncurated webly-supervised data. We download roughly ten times larger than what is used in alternative approaches (detailed in Appendix C). One might wonder: why not just scale up the datasets used by other methods? Retrieval-augmented techniques such as SuS-X (Udandarao et al., 2023) and Neural Priming—our (Wallingford et al., 2023) closest competitors performance-wise—experience stagnation or even a

Table 3: **Comparison with Name-Only Classification Techniques with ResNet50:** When comparing with existing state-of-the-art name-only classification techniques, we show that our method outperforms those methods by margins ranging from 2% to 25%.

| Type | Method | Model | Birdsnap | Aircraft | Flowers | Pets | Cars | DTD |
|------|--------|-------|----------|----------|---------|------|------|-----|
| Data-Free | CLIP-ZS (Radford et al., 2021) | CLIP | 32.6 | 19.3 | 65.9 | 85.4 | 55.8 | 41.7 |
| | CaFo-ZS (Zhang et al., 2023) | CLIP | - | 17.3 | 66.1 | 85.8 | 55.6 | 50.3 |
| | CALIP (Guo et al., 2023) | CLIP | - | 17.8 | 66.4 | 86.2 | 56.3 | 42.4 |
| | CLIP-DN (Zhou et al., 2023) | CLIP | 31.2 | 17.4 | 63.3 | 81.9 | 56.6 | 41.2 |
| | CuPL (Pratt et al., 2023) | CLIP | 35.8 | 19.3 | 65.9 | 85.1 | 57.2 | 47.5 |
| | VisDesc (Menon & Vondrick, 2022) | CLIP | 35.7 | 16.3 | 65.4 | 82.4 | 54.8 | 42.0 |
| | SD-Clf (Li et al., 2023a) | SD-2.0 | - | 26.4 | 66.3 | 87.3 | - | - |
| Use-Data | GLIDE-Syn (He et al., 2022) | CLIP | 38.1 | 22.0 | 67.1 | 86.8 | 56.9 | 43.2 |
| | CaFo (Zhang et al., 2023) | CLIP | - | 21.1 | 66.5 | 87.5 | 58.5 | 50.2 |
| | SuS-X-LC (Udandarao et al., 2023) | CLIP | 38.5 | 21.1 | 67.1 | 86.6 | 57.3 | 50.6 |
| | SuS-X-SD (Udandarao et al., 2023) | CLIP | 37.1 | 19.5 | 67.7 | 85.3 | 57.2 | 49.2 |
| | C2C (Ours-Linear Probe) | CLIP | 48.1 (+9.6) | 44.0 (+22.0) | 82.0 (+14.3) | 88.1 (+0.6) | 71.3 (+12.8) | 57.1 (+6.5) |
| | C2C (Ours-MLP Adapter) | CLIP | 46.6 (+8.1) | 48.9 (+26.9) | 84.8 (+17.1) | 89.4 (+1.9) | 72.6 (+14.1) | 57.6 (+7.0) |
| | C2C (Ours-Linear Probe) | MocoV3 | 56.1 (+17.6) | 57.5 (+35.5) | 85.7 (+18.0) | 91.7 (+4.2) | 62.1 (+3.6) | 54.6 (+4.0) |
| | C2C (Ours-MLP Adapter) | MocoV3 | 53.7 (+15.2) | 65.5 (+43.5) | 87.1 (+19.4) | 92.8 (+5.3) | 66.8 (+8.3) | 55.8 (+5.2) |

decline in results when expanded to 100 samples per class, as illustrated in Figure 6 of Udandarao et al. (2023) and discussed in Appendix B of Wallingford et al. (2023). Conversely, our method still achieves marked improvements in accuracy even as dataset sizes approach $500 - 750$ samples per class, as previously highlighted in Table 2. Alternative dataset generation methods, like Diffusion models (He et al., 2022; Zhang et al., 2023), come with a significant computational cost, yet they do not surpass retrieval methods such as LAION-5B in performance (Udandarao et al., 2023; Burg et al., 2023). To provide some context, producing a dataset equivalent in size to ours ($\sim 150$K samples) using generative techniques like stable-diffusion demands a staggering 32 hours of computation on an 8 A100 GPUs. In contrast, our approach collects the same dataset in around 15 minutes using a basic CPU machine.

## 5 CONTINUAL WEBLY-SUPERVISED LEARNING

③ Building upon our prior observations regarding the efficiency of collecting webly-supervised data and its effectiveness for *name-only* classification, we now test this approach in the context of *continual name-only* classification. Within this framework, the learner is solely provided with category names, and potentially descriptions, necessitating the continuous and streamlined construction of data and updation of the classifier. To assess the robustness and adaptability of our approach, we subject it to a diverse range of data streams encountered in various continual learning scenarios, namely: (i) *class-incremental:* the incremental addition of classes, (ii) *domain-incremental:* incremental adaptation to known domain shifts, and (iii) *time-incremental:* the gradual incorporation of new data over time. The subsequent subsection presents a comprehensive overview of the experimental setup and the corresponding results obtained from these three scenarios.

### 5.1 EXPERIMENTAL DETAILS

**Datasets:** We assess the effectiveness of our uncurated webly-supervised data in three different continual learning (CL) scenarios. For each scenario, we compare the performance of our downloaded data with manually annotated data. This evaluation setup aligns with the traditional CL paradigm, where labeled training data is revealed sequentially in the data stream. It is worth noting that methods with access to manually annotated data naturally have an inherent advantage. In principle, manually annotated data serves as a soft upper bound to our webly supervised approach. However, our primary goal is to determine to what extent web-supervised datasets can bridge this performance gap, with extreme limits of $< 1$ hour and cost $<\$15$ on AWS servers. Our experiments focus on the following three CL setups:

*Class-Incremental:* In this setting, we use CIFAR100, which is partitioned into ten timesteps, where at each timestep ten new class categories are introduced. CIFAR100 exhibits a notable domain gap due to its samples being old, centered, and downscaled to 32x32 pixels. To match this resolution, we downscale our images to 32x32 as well. The queries provided in this case simply consist of the class names for all previously encountered classes.

*Domain-Incremental:* In this setting, we use PACS (Li et al., 2017b) dataset, which comprises four timesteps and is suitable for the domain-incremental setup. Each timestep introduces new domains, namely Photos, Art, Cartoon, and Sketches. The primary challenge here lies in adapting to the distinct visual styles associated with each domain. The queries in this case are composed of a combination of class names and the names of the visual domains.

Table 4: **Linear Probe Performance in Continual Learning Scenarios (Avg. Acc. ↑).** Our uncurated webly-supervised data achieves average accuracy close to manually annotated (MA) datasets in a continual learning context with relatively small performance gaps.

| Eval Dataset | Split-CIFAR100 | Split-PACS | CLEAR10 |
|---|---|---|---|
| MA Data | 43.2% | 82.8% | 70.0% |
| C2C (Ours) | 38.7% ( -4.5%) | 80.8% ( -2.0%) | 65.3% ( -4.7%) |
| C2C (Top-20/engine/class) | 39.2% ( -4.0%) | 79.9% ( -2.9%) | 62.0% ( -8.0%) |
| C2C (Top-50/engine/class) | 39.5% ( -3.7%) | 78.6% ( -4.2%) | 60.8% ( -9.2%) |

*Time-Incremental:* In this setting, we use CLEAR10 (Lin et al., 2021) dataset, a recently popular CL dataset, by incorporating timestamps from the CLEAR benchmark into web queries[5]. Our web queries for categories are consistent across timesteps, however, samples are filtered by timestamp to match CLEAR time categorization. Here we only use Flickr as it supports timestamped querying.

**Optimizing Data Collection.** To optimize the creation of our webly-supervised datasets while adhering to time constraints, we conduct additional experiments involving the retrieval of only the top-$k$ most relevant samples per search engine. Specifically, we explore two settings: $k = 20$ and $k = 50$. This approach significantly diminishes the cost and time associated with querying the web and feature extraction processes.

**Training Models.** We note that we do not restrict the storage of past samples, unlike previous literature as download links largely remain accessible. If some download link expires then we do not that sample, allowing realistic privacy evaluation. However, we note that no links expired in the duration of our study and only a small fraction ($<5\%$) of links of CLOC dataset collected until 2014 have become invalid until today. Hence, we follow the constraints specified in Prabhu et al. (2023a) limiting the computational budgets and without storage constraints. We train a linear probe under varying Linear probing results are compared to NCM (Mensink et al., 2013; Janson et al., 2022) and KNN (Malkov & Yashunin, 2018; Prabhu et al., 2023b) classifiers. We use a ResNet50 MoCoV3 backbone for all experiments since SSL training has been shown to help in CL tasks (Gallardo et al., 2021). For linear probing, we use the same optimization parameters provided earlier except that we constrain the iterations according to our compute budgets $\mathcal{C}_t$. For more details about the computational budgets please refer to Appendix D. We set $k = 1$ and use cosine distance for KNN. During the training process, we implement experience replay and utilize class-balanced sampling to select training batches from the previously collected samples.

**Metrics.** We compute the average incremental accuracy for all three settings (Rebuffi et al., 2017). Briefly, we compute the accuracy on the available test set after finishing training of each timestep, which gives us a graph of accuracies over time. The average incremental accuracy is the aggregate measure of these incremental accuracies, which gives average performance of the method over time.

## 5.2 RESULTS

We evaluate the efficacy of our uncurated web data in the context of continual name-only learning and compare the results with manually annotated datasets in various scenarios. Linear probing results are presented in Table 4, while additional results for NCM and KNN can be found in the Appendix. Despite having access solely to class/category names, our uncurated webly-supervised data achieves accuracies that are close to training on the manually annotated datasets. We note that the performance on manually annotated datasets serves as an upper bound and not a fair comparison as they require expensive curation process, they are well-aligned with the test distribution as both sets are created from the same sampling of data. The performance gap between them is small, ranging from 2-5%, with the exception of CLEAR10 where it reaches 5-10%. In Table 6, we also consider the time required for querying and downloading our uncurated continual webly-supervised data. Remarkably, we are able to generate the web data within minutes, instead of days, across a variety of continual learning scenarios allowing more budget for computational resources. All experiments were completed in $< 1$ hour and cost $<\$15$ on AWS servers. This approach delivers both a performance comparable to manually annotated datasets and significantly reduces associated expenses, which typically exceed $4500.

**Understanding the Performance Gap:** While our webly-supervised dataset (C2C) has shown promise, a performance discrepancy exists when compared to the manually annotated data (MA Data) . This performance lag, is only slightly behind the ideal scenario of using in-distribution annotated data. The natural question that arises is: Why cannot we bridge this performance gap and

---

[5]Timestamps from: `https://github.com/linzhiqiu/continual-learning/blob/main/clear_10_time.json`

Table 5: **Comparing Last-Layer Based Continual Learning Approaches in Name-Only Continual Learning (Avg. Acc. ↑).** We evaluate the average accuracy of various continual learning methods in a name-only continual learning scenario with constrained computational resources. Surprisingly, KNN achieves superior results compared to linear probing, even while operating within a lower computational budget than the "tight" setting.

| Classifier | Budget | Split-CIFAR100 | Split-PACS | CLEAR10 |
|---|---|---|---|---|
| C2C-NCM | <Tight | 48.9% | 77.4% | 60.7% |
| C2C-Linear | Tight | 31.9% | 75.8% | 56.1% |
| C2C-Linear | Normal | 38.7% | 80.8% | 65.3% |
| C2C-Linear | Relaxed | 49.9% | 84.2% | 71.6% |
| C2C-KNN | <Tight | 59.8% (9.9%) | 89.5% (5.3%) | 81.2% ( 9.6%) |

Table 6: **(Left) Time Benchmark (in minutes):** Evaluation of the approximate total time required for continual learning across all time steps using a linear classifier. Our pipeline demonstrates exceptional time efficiency compared to manual sample annotation. **(Right) EvoTrend Results:** Comparative performance analysis of Linear Probing on C2C using MoCoV3 ResNet50 against zero-shot CLIP with ResNet50, indicating improved performance.

| | Benchmarks | | | | Method | Budget | Avg Acc. (↑) |
|---|---|---|---|---|---|---|---|
| Components | Split-CIFAR100 | Split-PACS | CLEAR10 | | ZS-CLIP RN50 | - | 62.9% |
| Download Images | ∼15 mins | ∼5 mins | ∼14 mins | | C2C-Linear | Tight | 57.7% ( -5.2%) |
| Extract Features | ∼15 mins | ∼6 mins | ∼10 mins | | C2C-Linear | Normal | 69.0% (+6.1%) |
| Model Training | ∼2 mins | ∼0.5 min | ∼1 min | | C2C-Linear | Relaxed | 72.8% (+9.9%) |
| Overall | ∼ 32 mins | ∼ 12 mins | ∼ 25 mins | | C2C-NCM | <Tight | 65.7% (+2.8%) |
| | | | | | C2C-KNN | <Tight | 78.6% (+15.7%) |

possibly exceed it, as observed in Section 4? Two primary distinctions from Section 4 can explain this: **(i)** The current training operates within a limited computational budgets, and **(ii)** The size difference between our webly-supervised continual datasets and manually annotated datasets has notably shrunk, transitioning from a substantial $30 - 100\times$ difference to a mere $2 - 3\times$. It is important to note that in Section 4, when we match the size of the manually annotated datasets by considering only the top-20 web queries, we observe a similar gap to that in this section. Nevertheless, the improvements in speed and reduction in annotation costs significantly outweigh this gap.

Firstly, in the case of PACS, a significant domain gap arises between web sketches which refer to line-drawings and manually annotated sketches which refer to quick-draws. This domain shift results in a performance gap, which is challenging to bridge with the inclusion of additional sketch data from the internet. Second, in CIFAR100, images are carefully selected and often do not reflect real-world data streams. Specifically, they consist of older, centered, and downsampled images, which strongly contrast with the dynamic and varied nature of web data harnessed in our approach. This difference highlights the importance of considering more realistic data streams, over hand-picked and potentially unrepresentative datasets. Lastly, in the context of CLEAR10, our analysis uncovers data collection inaccuracies, particularly in the bus class. While our web datasets consist of images depicting the exterior of buses, the manually annotated CLEAR10 dataset primarily includes interior images of buses in the train/test set. Given that the bus class constitutes one out of ten classes, our webly-supervised approach directly incurs a 10% performance loss illustrated in Figure 4 in Appendix D. We delve deeper into this domain gap issue in Appendix D.

### 5.3 EvoTrends: The First Continual Name-Only Classification Benchmark

To demonstrate the adaptability, speed, and effectiveness of our webly-supervised approach to continual name-only classification, we introduce a novel dataset titled "EvoTrends" (depicted in the Appendix D). Instead of resorting to synthetic methods that produce class-incremental streams based on arbitrarily chosen classes, EvoTrends offers a more natural scenario. Spanning two decades (2000 to 2020), it chronicles a sequence of trending products that have risen in popularity, from the PlayStation 2 in 2000 to face masks in 2020. This dataset spans 21 unique timesteps and comprises 39 different classes. EvoTrends, presents a real-world class-incremental challenge where new "trend" classes emerge naturally over time. With our approach, and without requiring any specific alterations to our method, we query and download webly-supervised data for each timestep. Within minutes, our model could continually adapt and effectively categorize these evolving trends, establishing its viability for applications demanding rapid adaptation to unlabeled, continuously updated data or development of new classifiers. The dataset was divided into 36,236 training samples and 11,317 testing samples, with the test set cleaned using automated deduplication and anomaly detection pipelines.

**Results.** Preliminary evaluations on the EvoTrends dataset have yielded promising results. Our method was compared to the zero-shot CLIP. It is noteworthy that CLIP may have already been

trained on several EvoTrends classes during its pretraining phase, as illustrated in Table 6 (right). Impressively, our model, while using a ResNet50 MoCoV3 and budgeting compute, outperformed CLIP-RN50 by 6.1%. This further underscores the potency of our method. Consistent with previous sections, KNN stands out as the most proficient performer within our computational parameters.

## 6 CONCLUSION

In conclusion, the traditional reliance on costly and time-intensive manual annotation for Continual Learning is increasingly becoming infeasible, especially in rapidly changing environments. Our exploration into the *name-only continual learning* scenario presents a challenging and novel CL setting. Our proposed solution which leverages uncurated webly-supervised data (C2C), does not only manage to reduce the annotation costs significantly but also maintain a high level of classifier performance. Addressing key questions regarding the reliability, comparison to state-of-the-art name-only classification, and adaptability across various continual learning settings, we demonstrate that webly-supervised data can offer a robust and efficient alternative to manual data curation. We also present *EvoTrends* a dataset that highlights our capability to swiftly adapt to emerging trends using web-supervision. We believe our work can lead to lots of avenues for improvements:

*(i) Extending the Setting to Test-Time Adaptation using the Web:* An interesting extension is to incorporate unlabeled test images together with their corresponding categories, enabling test-time adaptation. This allows retrieval-based refinement of web search results (Oquab et al., 2023). This additionally allows gathering webly-supervised data close to test samples using reverse-image search-based retrieval on the web.

*(ii) Online Continual Learning:* Our method is not yet equipped to handle the challenges posed by online continual learning streams, where we have to rapidly adapt to incoming stream of images. It may worsen the issues related to data correlations due to repeated scraping of trending images from various sources (Hammoud et al., 2023). Cleaning and analyzing name-only continual learning scenarios in a online fashion, with rapid distribution shifts, presents unique challenges, especially without labels, which are currently beyond the scope of our work.

*(iii) Data Collection Limitations:* Although our approach is effective in many scenarios, it may not be suitable for a significant portion of niche domain datasets, particularly in fields like medicine where manual collection remains the primary source due to data sharing constraints. It is important to verify and acknowledge the limitations of the name-only classification setting itself.

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

## A  CONNECTIONS TO PAST LITERATURE IN WEBLY SUPERVISED LEARNING

Webly-supervised learning has been extensively explored in the past two decades. We summarize a few directions in this section.

**Weakly/Noisy-Label Supervised Learning**: Early works attempt to leverage alternative forms of input signals, such as predicting words (Joulin et al., 2016) or n-grams (Li et al., 2017a). Subsequent work (Mahajan et al., 2018; Singh et al., 2022; Beal et al., 2022; Ghadiyaram et al., 2019) in this direction focused on significantly increasing the size of pretraining datasets, often by incorporating hashtags. This expansion of pretraining data was aimed at improving model performance. More recently, there has been a surge in interest in vision-language pretraining (Radford et al., 2021; Jia et al., 2021; Cherti et al., 2023), particularly with alt-text training, which has gained popularity due to its out-of-the-box, zero-shot capabilities. However, it is important to note that these methods often grapple with the issues of poor ground truth quality leading to a noisy training signal. Interestingly, they can be matched in performance by self-supervised learning methods (Oquab et al., 2023) which do not rely on any annotations and often involve fewer images in their training process.

**Augmenting Training Datasets with Internet**. An promising alternative is to augment training datasets with web images Zheng et al. (2020). Recently, (Oquab et al., 2023) obtained highly transferable representations by augmenting training dataset with high quality web images by similarity-search for self-supervised pretraining. Additionally, parallel work (Li et al., 2023b) explored targeted pretraining, making that step highly efficient given a downstream task. However, our setting differs from these works as they focus on pretraining and have supplementary train data available, we focus on the label-only setting with no reference train set in the downstream task.

**Name-only Classification.** Image search engines traditionally provide highly relevant results, using text, linked images and user query based recommendation engines. Hence, researchers have long-since used internet in the name-only classification setting as a form of supervision (Fergus et al., 2005; Vijayanarasimhan & Grauman, 2008; Schroff et al., 2010). Several datasets including Webvision (Li et al., 2017c), JFT-300M (Sun et al., 2017), Instagram-1B (Mahajan et al., 2018) have been introduced for pretraining. Several works have specifically targeted fine-grained classification (Krause et al., 2016; Sun et al., 2021) creating very large scale pretraining datasets for fine-grained classification. Recent approaches have primarily focused on open-ended large-vocabulary visual category learning (Kamath et al., 2022). Past works have diversified this to object detection (Chen & Gupta, 2015) and object segmentation (Shen et al., 2018; Jin et al., 2017; Sun et al., 2020) with tackling challenges of the weak category supervision. We do not focus on the pretraining task, where self-supervised learning has achieved dominance. Instead, we focus on building targeted representations for downstream classification tasks.

**Continual Webly-Supervised Learning.** This line of research which attempts to learn datasets continually (Li & Fei-Fei, 2010) is much sparser, with classical works (Divvala et al., 2014; Chen et al., 2013) attempting to learn exhaustively about visual categories. In contrast, we focus on more recent formulation of the continual learning problem with a targeted set of categories, computationally budgeted learning when given class categories.

**Active Learning.** An interesting avenue of research is actively labeling portions of dataset with humans-in-the-loop allowing high quality annotations within limited time. A primary bottleneck here is the computationally expensiveness in scaling these approaches to large datasets, tackled in (Coleman et al., 2019; Prabhu et al., 2019). However, their primary quality is they allow learning rare, underspecified concepts (Coleman et al., 2022; Hayes et al., 2022; Stretcu et al., 2023) in a continual fashion (Mundt et al., 2023; Munagala et al., 2022). However, these methods are complementary to name-only classification approaches which generate the unlabeled pool of images for these approaches to select informative samples from.

# B  APPENDIX: HOW RELIABLE IS UNCURATED WEBLY-SUPERVISED DATA?

**How Big Is Our Dataset?** We present the dataset set statistics of both the manually curated and the web data in Table 7. The scale of the uncurated webly-supervised data is significantly bigger than the manually annotated ones. Additionally, cleaning the web data using automated removal of noisy samples and deduplication using DinoV2 ViT-G (Oquab et al., 2023) features could significantly reduce the size of the web data, indicating severe duplication amongst search engines.

| | **Datasets** | | | | |
| Training Dataset | FGVC Aircraft | Flowers102 | OxfordIIITPets | Stanford Cars | BirdSnap |
|---|---|---|---|---|---|
| *Size Relative to Ground Truth Train Sets* | | | | | |
| MA Data | 3.3K | 1.0K | 3.7K | 8.1K | 42K |
| C2C (Ours) | 158K (48x) | 155K (155x) | 53K (15x) | 184K (23x) | 557K (13x) |
| *Ablating Dataset Cleaning* | | | | | |
| Before Cleaning | 158K | 155K | 53K | 184K | 557K |
| Duplicates | 57K | 51K | 10K | 57K | - |
| Noisy Samples | 15K | 23K | 2.3K | 24K | - |
| After Cleaning | 90K (0.57x) | 97K (0.62x) | 40K (0.75x) | 111K (0.60x) | - |
| *Ablating Search Engines* | | | | | |
| All | 158K | 155K | 53K | 184K | 557K |
| Google Only | 30K | 29K | 11K | 60K (1.3x) | 142K |
| Bing Only | 29K | 30K | 8K | 47K | 104K |
| DuckDuckGo Only | 29K | 32K | 8K | 55K | 166K (1.1x) |
| Flickr Only | 70K (2.3x) | 65K (2.0x) | 26K (2.4x) | 21K | 145K |

Table 7: **Dataset Statistics**. Our uncurated webly-supervised data is significantly larger in size compared to manually annotated (MA) datasets. Leveraging multiple engines could allow to query the web for samples enabling the creation of datasets up to $155\times$ larger than manually curated ones.

## B.1  ANALYSIS: WHY DOES UNCURATED WEB DATA OUTPERFORM MANUALLY ANNOTATED DATASETS?

We now present a comprehensive analysis of factors that we thoroughly examined but found to have no impact on our performance.

**1. Uncurated Webly-Supervised Data on Different Architectures and Training Procedures**

To investigate this aspect, we conducted an experiment employing three additional backbones: (a) a supervised ImageNet1K ResNet50, (b) a DeiT-III ViT-B/16 ImageNet21K, and (c) a weakly-supervised CLIP ResNet50 (Radford et al., 2021). The results of this experiment are summarized in Table 8. Our findings reveal that neither the architecture nor the training procedure is the underlying cause behind the observed performance. Remarkably, our web data consistently outperform manually annotated datasets across various classifiers and training procedures. In conclusion, our study consistently demonstrates comparable or superior performance to manually annotated datasets when considering different classifiers and training procedures.

**2. Influence of Specific Search Engines on Accuracy Improvement**

To investigate the impact of individual search engines on our overall performance, we compare the performance achieved by each of the four search engines individually, as presented in Table 9. Our analysis reveals that the collection of data from all four search engines, despite encountering challenges such as duplicate search results, consistently results in a 3-10% higher absolute accuracy improvement compared to using a single search engine. It is worth noting that Flickr consistently outperforms the other engines, while Google exhibits the lowest performance. In summary, the performance of our webly-supervised data cannot be solely attributed to a specific search engine. Each search engine contributes a distinct distribution of samples, and the combination of these distributions leads to enhanced generalization and improved performance.

**3. Does Cleaning Webly-supervised Data Help?** To clean our data, we perform removal of noisy samples along with deduplication using DinoV2 ViT-G features for reliability. We defined duplicates as samples with a cosine similarity higher than 0.99. We defined noisy samples as samples which

| Eval. | Dataset | Datasets | | | | |
|---|---|---|---|---|---|---|
| | | **FGVC Aircraft** | **Flowers102** | **OxfordIIITPets** | **Stanford Cars** | **BirdSnap** |
| | | *ResNet50 (Supervised)* | | | | |
| Linear Probing | MA Data | 31.7% | 79.9% | 92.1% | 48.8% | 50.4% |
| | C2C (Ours) | 44.2% (+12.5%) | 83.5% (+3.6%) | 92.9% (+0.8%) | 58.1% (+9.3%) | 56.2% (+5.8%) |
| MLP Adapter | MA Data | 35.5% | 78.8% | 92.8% | 45.6% | 49.2% |
| | C2C (Ours) | 47.3% (+11.8%) | 84.5% (+5.7%) | 93.8% (+1.0%) | 58.0% (+12.4%) | 54.5% (+5.3%) |
| | | *ViT-B/16 (DeiT-III)* | | | | |
| Linear Probing | MA Data | 49.0% | 98.0% | 94.3% | 64.5% | 73.7% |
| | C2C (Ours) | 63.5% (+14.5%) | 92.9% (-5.1%) | 94.8% (+0.5%) | 71.5% (+7.0%) | 73.4% (-0.3%) |
| MLP Adapter | MA Data | 48.9% | 98.3% | 94.3% | 65.7% | 72.9% |
| | C2C (Ours) | 67.4% (+18.5%) | 93.6% (-4.7%) | 95.0% (+0.7%) | 73.9% (+8.2%) | 75.4% (+2.5%) |
| | | *ViT-B/32 (CLIP)* | | | | |
| Linear Probing | MA Data | 40.5% | 91.2% | 89.5% | 80.8% | 56.1% |
| | C2C (Ours) | 51.7% (+11.2%) | 86.2% (-5.0%) | 91.8% (+2.3%) | 79.3% (-1.5%) | 58.4% (+2.3%) |
| MLP Adapter | MA Data | 44.8% | 88.9% | 90.9% | 79.9% | 55.3% |
| | C2C (Ours) | 57.8% (+13.0%) | 86.8% (-2.1%) | 92.6% (+1.7%) | 81.1% (+1.2%) | 57.1% (+1.8%) |
| | | *ResNet50-CLIP* | | | | |
| Linear Probing | MA Data | 31.9% | 80.5% | 81.8% | 69.2% | 44.7% |
| | C2C (Ours) | 44.0% (+12.1%) | 82.0% (+1.5%) | 88.1% (+6.3%) | 71.3% (+2.1%) | 48.1% (+3.4%) |
| MLP Adapter | MA Data | 34.3% | 78.3% | 85.7% | 69.0% | 45.2% |
| | C2C (Ours) | 48.9% (+14.6%) | 84.8% (+6.5%) | 89.4% (+3.7%) | 72.6% (+3.6%) | 46.6% (+1.4%) |

Table 8: **Impact of Architecture and Pretraining Algorithm.** Irrespective of the selected network architecture and pretraining algorithm, our uncurated webly-supervised data consistently exhibit superior performance over manually annotated datasets by substantial margins. The improvement is primarily attributed to the scale of web data.

| Eval. | Training Dataset | Datasets | | | | |
|---|---|---|---|---|---|---|
| | | **FGVC Aircraft** | **Flowers102** | **OxfordIIITPets** | **Stanford Cars** | **BirdSnap** |
| Linear Probe | All | 57.5% | 85.7% | 91.7% | 62.1% | 56.1% |
| | Google Only | 42.0% (-15.5%) | 75.7% (-10.0%) | 88.9% (-2.8%) | 63.2% (+1.1%) | 48.9% (-7.2%) |
| | Bing Only | 48.9% (-8.6%) | 81.3% (-4.4%) | 88.8% (-2.9%) | 55.7% (-6.4%) | 47.2% (-8.9%) |
| | DuckDuckGo Only | 47.4% (-10.1%) | 80.4% (-5.3%) | 88.8% (-2.9%) | 59.0% (-3.1%) | 46.7% (-9.4%) |
| | Flickr Only | 58.9% (+1.4%) | 82.5% (-3.2%) | 89.9% (-1.8%) | -% | 52.8% (-3.3%) |
| MLP-Adapter | All | 65.5% | 87.1% | 92.8% | 66.8% | 53.7% |
| | Google Only | 49.2% (-16.3%) | 78.3% (-8.8%) | 90.3% (-2.5%) | 65.6% (-1.2%) | 48.9% (-4.8%) |
| | Bing Only | 54.5% (-11.0%) | 81.7% (-5.4%) | 90.1% (-2.7%) | 60.3% (-6.5%) | 48.7% (-5.0%) |
| | DuckDuckGo Only | 54.0% (-11.5%) | 81.6% (-5.5%) | 90.1% (-2.7%) | 62.4% (-4.4%) | 47.5% (-6.2%) |
| | Flickr Only | 62.8% (-2.7%) | 83.0% (-4.1%) | 91.7% (-1.1%) | -% | 50.7% (-3.0%) |

Table 9: **Performance Across Search Engines.** We find that different search engines demonstrate varying levels of performance across different datasets, and their collective utilization yields superior results compared to any individual search engine. It is worth noting that Flickr encountered difficulties in querying and downloading images for certain classes in the Cars dataset, hence the performance for those classes is not reported.

have a cosine similarity lower than 0.2 for 50% or more samples in their category. We checked that our thresholds were reasonable by manual inspection of identified duplicate and noisy samples. In table 10, we investigate the impact of cleaning the noisy webly-supervised data. We find that cleaning our data with did not lead to substantial performance improvements, indicating that the main challenge may be the out-of-domain nature of web data rather than the inherent noise in the queried samples.

**4. Does Class Balancing Help?** In Table 10 we investigate the importance of class balanced sampling in our setup. We find that class-balanced sampling, did not impact performance much and only lead to drops in performance in a few cases. This suggests that further improvements in this direction, such as using long-tailed loss functions, might not yield significant improvements. Our

| | | Datasets | | | | |
|---|---|---|---|---|---|---|
| **Evaluation** | **Training Dataset** | **FGVC Aircraft** | **Flowers102** | **OxfordIIITPets** | **Stanford Cars** | **BirdSnap** |
| | | *Ablating Dataset Cleaning* | | | | |
| Linear Probing | Before Cleaning | 57.5% | 85.7% | 91.7% | 62.1% | 56.1% |
| | After Cleaning | 57.3% (-0.2%) | 83.3% (-2.4%) | 91.7% | 69.2% (+7.1%) | - |
| MLP Adapter | Before Cleaning | 65.5% | 87.1% | 92.8% | 66.8% | 53.7% |
| | After Cleaning | 64.6% (-0.9%) | 84.8% (-2.3%) | 93.1% (+0.3%) | 71.0% (+4.2%) | - |
| | | *Ablating Balanced Training* | | | | |
| Linear Probing | Class-Balanced | 57.5% | 85.7% | 91.7% | 62.1% | 56.1% |
| | Uniform | 61.2% (+3.7%) | 84.6% (-1.1%) | 91.8% (+0.1%) | 68.7% (+6.6%) | 56.8 %(+0.7%) |
| MLP Adapter | Class-Balanced | 65.5% | 87.1% | 92.8% | 66.8% | 53.7% |
| | Uniform | 66.2% (+0.7%) | 86.4% (-0.7%) | 92.6% (-0.2%) | 70.1% (+3.3%) | 53.8% (+0.1%) |

Table 10: **Effect of Architecture and Pretraining Algorithm.** We observe different search engines performing well across different datasets, with their combined data consistently outperforming any individual search engine.

investigation suggests that categories for which we can only collect a limited number of samples from the web are typically associated with noisy samples. Consequently, these categories do not contribute significantly to performance improvement.

## C  APPENDIX: COMPARISON WITH NAME-ONLY CLASSIFICATION STRATEGIES

We provide detailed descriptions of the compared approaches below:

1. **CALIP** (Guo et al., 2023): This approach aims to enhance the alignment between textual and visual features for improved similarity score estimation. It achieves this by utilizing textual class-prompts to generate attention maps that correlate better with the input images.

2. **CLIP-DN** (Zhou et al., 2023): Similar to CALIP, CLIP-DN focuses on enhancing the alignment between textual and visual features. It achieves this through test-time adaptation by estimating input normalization.

3. **CuPL** (Pratt et al., 2023): CuPL takes a different approach by improving prompts at a class-wise granularity across datasets. Leveraging GPT3, it generates enhanced prompts for each class.

4. **VisDesc** (Menon & Vondrick, 2022): VisDesc improves prompts by computing similarity with multiple textual descriptors that capture the visual characteristics of the target class.

5. **Glide-Syn** (He et al., 2022): Glide-Syn generates realistic synthetic images using the ALIGN model. It then uses classifier tuning, as proposed in (Wortsman et al., 2022), to classify these synthetic images.

6. **SuS-X-LC and SuS-X-SD** (Udandarao et al., 2023): SuS-LC and SuS-SD experiment with different techniques. SuS-LC retrieves images similar to the class-prompt from the LAION-5B dataset using a ViT-L model. SuS-SD, on the other hand, generates images based on a class-description using a stable diffusion model. These images are then utilized in combination with an adapter module called Tip-X for inference.

7. **SD-Classifier** (Li et al., 2023a): SD-Classifier employs a diffusion-classifier framework to fine-tune the text conditioning **c** for each class. It predicts the noise added to the input image that maximizes the likelihood of the true label.

8. **CaFo** (Zhang et al., 2023): CaFo leverage GPT-3 to produce textual inputs for CLIP and for querying a DALL-E model in the non-zs variant. The DALL-E model then generates a classification dataset with 16 samples per class, using a combination of CLIP and DINO features.

9. **Retrieval+SSL** (Li et al., 2023b; Wallingford et al., 2023): This implements a self-supervised finetuning step with the base backbone before classification with label information. However, this is a 1-shot internet explorer algorithm, which might not be faithful to the internet explorer idea as repeated exploration and refinement is not possible in a name-only setting with no access to training dataset.

10. **Neural Priming** (Wallingford et al., 2023): Similar to SuS-X-LC, it retrieves samples relevant to the downstream task from LAION2B. However, instead of relying on semantic search on CLIP features, they adopt a search using language for fast initial filtering and image search for accurate retrieval. Additionally, they retrieve samples only from LAION2B, their pretraining set, unlike past works and use a weighted combination of Nearest Class Mean (NCM) and Linear probe for classification.

**Additional Results.** Table 12 shows the number of samples used in prior-art compared to our webly-supervised data. Our webly-supervised data uses up-to $155\times$ more samples than previous methods. Table 11 shows similar results and trends as presented in the manuscript for CLIP-ResNet50 but for CLIP-ViTB/16. Our method outperforms prior art by large margins with the exception of Stanford Cars where Neural Priming outperforms our approach.

| Type | Method | Pretraining | Birdsnap | Aircraft | Flowers-102 | Pets | Cars | DTD |
|---|---|---|---|---|---|---|---|---|
| Data-Free | CLIP-ZS (Radford et al., 2021) | CLIP | 39.1 | 27.1 | 70.4 | 88.9 | 65.6 | 46.0 |
| | CLIP-ZS | OpenCLIP-2B | - | 25.9 | 71.7 | 90.2 | 87.4 | - |
| | CuPL (Pratt et al., 2023) | OpenCLIP-2B | - | 29.6 | 72.3 | 91.2 | 88.6 | - |
| | VisDesc (Menon & Vondrick, 2022) | CLIP | - | - | - | 86.9 | - | 45.6 |
| | VisDesc (Menon & Vondrick, 2022) | OpenCLIP-2B | - | 28.5 | 72.1 | 90.2 | 87.9 | - |
| Use-Data | GLIDE-Syn (He et al., 2022) | CLIP | 46.8 | 30.8 | 72.6 | 90.4 | 66.9 | 44.9 |
| | SuS-LC (Udandarao et al., 2023) | CLIP | 47.7 | 30.5 | 73.8 | 91.6 | 65.9 | 55.3 |
| | SuS-SD (Udandarao et al., 2023) | CLIP | 45.5 | 28.7 | 73.1 | 90.6 | 66.1 | 54.6 |
| | Retrieval+SSL* (*Li et al.*, 2023b) | OpenCLIP-2B | - | 26.2 | 72.1 | 90.4 | 88.0 | - |
| | Priming (Wallingford et al., 2023) | OpenCLIP-2B | - | 33.0 | 79.8 | 91.9 | 89.3 | - |
| | Priming+CuPL (Wallingford et al., 2023) | OpenCLIP-2B | - | 36.0 | 80.0 | 91.9 | 90.2 | - |
| | C2C (Ours-Linear) | CLIP | 62.7 (+15.0) | 62.0 (+26.0) | 87.8 (+7.8) | 94.1 (+2.2) | 83.8 (-6.2) | 61.0 (+5.7) |
| | C2C (Ours- MLP Adapter) | CLIP | 64.4 (+16.7) | 66.2 (+30.2) | 89.9 (+9.9) | 94.5 (+2.6) | 85.5 (-4.7) | 61.8 (+6.5) |
| | C2C (Ours-Linear) | DeIT-III (Im21K) | 73.4 (+25.7) | 63.5 (+27.5) | 92.9 (+12.9) | 94.8 (+2.9) | 71.5 (-18.7) | 57.6 (+2.3) |
| | C2C (Ours-MLP Adapter) | DeIT-III (Im21K) | 75.4 (+27.7) | 67.4 (+31.4) | 93.6 (+13.6) | 95.0 (+3.1) | 73.9 (-16.3) | 60.4 (+5.1) |

Table 11: **ViTB/16: Comparison with Other Approaches for Name-Only Classification** Our uncurated webly-supervised data consistently outperform existing name-only classification techniques, with the exception of the Stanford Cars dataset where Priming outperforms our approach. * indicates that the method significantly differs from the cited work. However, it performs SSL finetuning with retrieved samples, emulating a 1-step Internet-explorer algorithm. Further exploration of LAION2B is not possible here due to the lack of a training set.

| | Datasets | | | | |
|---|---|---|---|---|---|
| Training Dataset | FGVC Aircraft | Flowers102 | OxfordIIITPets | Stanford Cars | BirdSnap |
| *Size Relative to Ground Truth Train Sets* | | | | | |
| CaFo (Zhang et al., 2023) | 1.6K | 1.6K | 0.6K | 3.1K | 8.0K |
| GLIDE-Syn (He et al., 2022) | 3.3K | 1.0K | 3.7K | 8.1K | 42K |
| SuS-X (Udandarao et al., 2023) | 7.9K | 3.2K | 2.6K | 1K | 39K |
| C2C (Ours) | 158K (48x) | 155K (155x) | 53K (15x) | 184K (23x) | 557K (13x) |

Table 12: **Dataset Statistics**. Our uncurated webly-supervised data is significantly larger in size compared to other approaches for acquiring a training set. Leveraging multiple engines could allow to query the web for samples enabling the creation of datasets up to $15-155\times$ larger than alternative approaches for creating a training set in a name-only classification setting.

# D  APPENDIX: CONTINUAL WEBLY SUPERVISED LEARNING

## D.1  EVOTRENDS

**EvoTrends:** Figure 2 visually depicts the construction of our dataset, EvoTrends. This dataset captures the trends of products over a span of 21 years, starting from the year 2000 and continuing until 2020. Each year showcases a distinct trending product, such as the PlayStation 2 in 2000 and N95 Masks in 2020. What sets EvoTrends apart is its truly class-incremental setup, where the learner must quickly adapt to emerging concepts and categories, as opposed to the conventional class-incremental setup where classes are incremented arbitrarily.

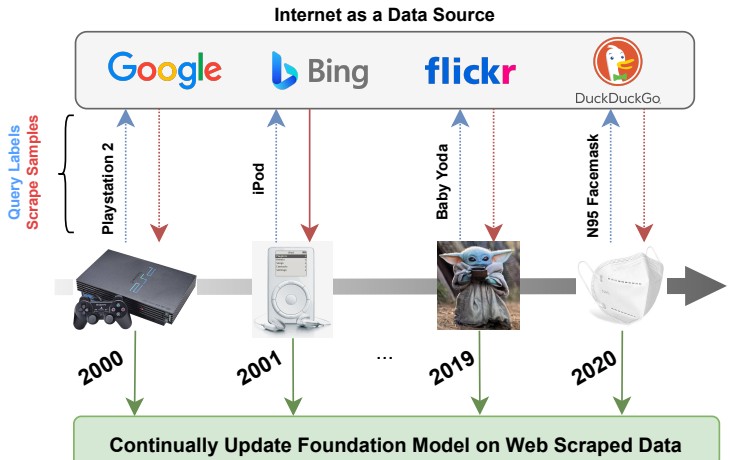

Figure 2: **EvoTrends: A Dynamic Dataset Reflecting Real-world Trends.** This illustration showcases the dataset we have curated using internet sources. EvoTrends consists of 21 timesteps, spanning the years 2000 to 2020. Each timestep presents the most trending products of the respective year, challenging the learner to adapt to these evolving trends. Unlike artificial scenarios, this dataset accurately reflects a real class-incremental setting, where classes emerge based on actual trends observed in the world.

## D.2  COMPUTATIONAL BUDGETS

**Computational Budgets.** In Figure 3 we present a visual representation of the impact of different computational budgets on the training epochs. Each row in the figure corresponds to one of the computational budgets: tight, normal, and relaxed, in sequential order. Specifically, the normal budget is selected to allow the manually annotated datasets to undergo one epoch of training during the initial time step, as depicted by the green plot in the left side figures. Comparatively, the tight budget, represented by the blue plot, is half the size of the normal budget, while the relaxed budget, indicated by the yellow plot, is four times the size of the normal budget. As the webly-supervised data surpasses the manually curated datasets in terms of size, the number of epochs they undergo within each of the three budget regimes decreases. This is due to the increasing amount of data presented at each time step, resulting in a decay in the effective number of epochs over time.

**Dataset Size Comparison and Computational Budget Analysis.** We experiment with different levels of computational constraints and present the results in Appendix Table 13. We find that accuracy predictably improves with increasing computational budgets, while the gap between our webly-supervised approach and manually annotated datasets remains relatively similar.

| Eval. | Training Dataset | Avg. Acc. on Benchmarks | | |
|---|---|---|---|---|
| | | Split-PACS | Split-CIFAR100 | CLEAR10 |
| Tight | GT Data | 77.1% | 35.9% | 62.1% |
| | C2C (Ours) | 75.8% | 31.9% | 56.1% |
| | C2C (Top-20/engine/class) | 73.3% | 31.1% | 52.3% |
| | C2C (Top-50/engine/class) | 72.8% | 31.3% | 56.5% |
| Relaxed | GT Data | 88.6% | 57.7% | 81.8% |
| | C2C (Ours) | 84.2% | 49.9% | 71.6% |
| | C2C (Top-20/engine/class) | 82.9% | 48.0% | 66.9% |
| | C2C (Top-50/engine/class) | 84.1% | 50.0% | 70.0% |

Table 13: **Consistency of Linear Probing Performance across Computational Budgets.** Our findings reveal a remarkable consistency in the results in terms of performance gap across different computational budgets.

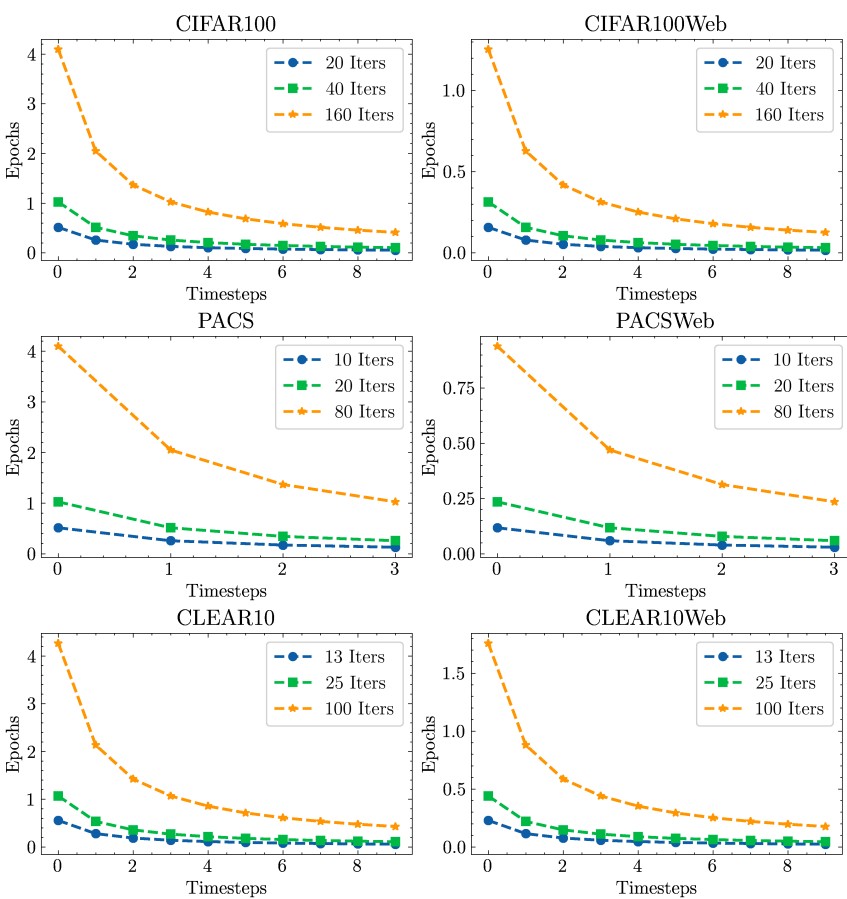

Figure 3: **Effective Training Epochs Per Time Step.** Each row represents one of the computational budgets: tight, normal, and relaxed in sequential order. The normal budget is carefully chosen to allow the manually annotated datasets to undergo one epoch of training during the initial timestep (depicted by the green plot in the left side figures). The tight budget (blue) is half the budget of normal, while the relaxed budget (yellow) is four times the budget of normal. As the webly-supervised data surpasses the manually curated datasets in size, they undergo fewer epochs within each of the three budget regimes. At each timestep more data is presented, hence the effective number of epochs decays with time.

## D.3 INVESTIGATING PERFORMANCE GAPS BETWEEN OUR WEBLY-SUPERVISED DATA AND MA DATA

In this section, we inspect the reasons for the performance gap between using our webly-supervised data compared to the manually annotated data in the *continual name-only* setup.

**Domain Gaps.**

**CLEAR10.** In Table 14 we show class-wise performance analysis of our approach on CLEAR10 dataset. This analysis reveals a noticeable decrease in accuracy for the bus class. Upon further inspection, we found that this decline can be attributed to the disparity between the CLEAR10 test set, which contains images of the bus's interior, and our webly-supervised data, which predominantly consists of images depicting the bus's exterior. Consequently, the accuracy for our bus class is significantly lower. For a visual representation of this discrepancy, please refer to Figure 4.

| Query Category | Accuracy |
|---|---|
| Baseball | 81.4% |
| Bus | 19.5% |
| Camera | 86.0% |
| Cosplay | 96.2% |
| Dress | 69.6% |
| Hockey | 88.2% |
| Laptop | 95.8% |
| Racing | 77.6% |
| Soccer | 65.2% |
| Sweater | 89.4% |

Table 14: **CLEAR10 Class-wise Performance.** The analysis of class-wise performance in CLEAR10 highlights a significant decline in accuracy specifically for the bus class. This discrepancy can be attributed to the fact that CLEAR10 test set includes images of the "interior" of the bus, whereas our webly-supervised data primarily consists of images depicting the "exterior" of the bus. As a result, the accuracy for our bus class is notably low. Please refer to Figure 4 for a visual representation of this disparity.

**PACS.** In Table 15 we show an analysis of domain-specific accuracy in Continual PACS which reveals important insights. Our domain-incremental setup involves a dataset comprising four distinct domains: Photo, Art, Cartoon, and Sketch. It is crucial to acknowledge the substantial disparity between the sketches in PACS and our webly-supervised data. While sketches in PACS are characterized as quick drawings, our webly-supervised data encompasses more detailed sketches and line drawings. This domain shift has a significant impact on the performance of our approach, particularly in the Sketch domain, where we observe a notable drop in accuracy. To visually comprehend this domain shift, please refer to Figure 5.

| Dataset | Photo | Art | Cartoon | Sketch |
|---|---|---|---|---|
| MA Data | 99.4% | 91.6% | 85.9% | 77.4% |
| C2C (Ours) | 96.4% -3.0% | 90.7% -0.9% | 83.0% -2.9% | 66.3% -11.1% |

Table 15: **Domain-specific Accuracy in Continual PACS.** Our domain-incremental setup incorporates a dataset consisting of four domains: Photo, Art, Cartoon, and Sketch. It is important to note that there is a notable disparity between the sketches in PACS, which are characterized as quick drawings, and the sketches in our webly-supervised data, which encompass more detailed sketches and line drawings. This domain shift results in a significant performance drop for our approach in the Sketch domain. This domain shift can be observed in Figure 5.

**CIFAR100.** In Figure 6 we present a comparison between manually annotated CIFAR100 and our webly-supervised version of it which reveals a substantial domain gap. The upper row showcases the $32 \times 32$ samples obtained from CIFAR100, while the second row displays images collected through our webly-supervised approach. It is evident that the CIFAR100 images exhibit characteristics such as low quality, centered composition, and an outdated nature. In contrast, the samples

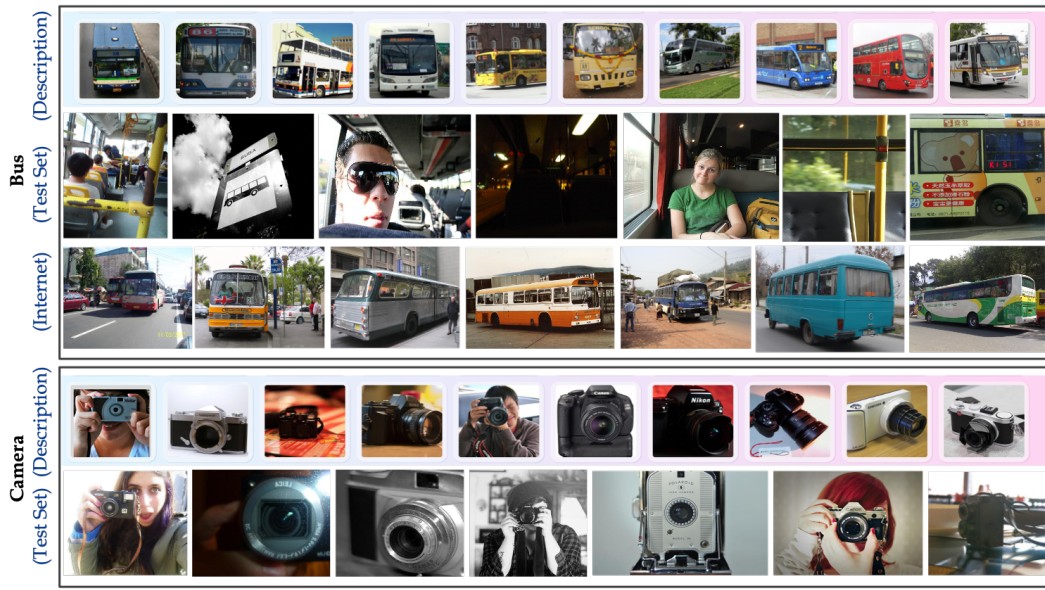

Figure 4: **Domain Gap in CLEAR10 Dataset.** In the CLEAR10 dataset, the original paper describes buses as the "exterior" of the bus, while the test set predominantly consists of images showcasing the "interior" of the bus. In contrast, our web-collected data primarily comprises "exterior" photos of the buses. Considering the inherent dissimilarity between our buses and the test set, this justifies the 10% performance gap observed when using our webly-supervised data compared to manually annotated datasets. It is important to note that this discrepancy does not apply to other classes within CLEAR10, as demonstrated by the camera class for reference.

| Eval. Dataset | Datasets | | |
|---|---|---|---|
| | **Split-CIFAR100** | **Split-PACS** | **CLEAR10** |
| *Size Relative to Ground Truth Train Sets* | | | |
| MA Data | 50K | 8K | 30K |
| C2C (Ours) | 163K (3.3x) | 44K (5.5x) | 73K (2.4x) |
| C2C (Top 20/engine/class) | 8K (0.16x) | 2.2K (0.3x) | 2K (0.07x) |
| C2C (Top 50/engine/class) | 20K (0.4x) | 5.6K (0.7x) | 5K (0.17x) |

Table 16: **Statistics of Continual Datasets**. In contrast to the datasets collected in the *name-only* classification setup, the scale of the datasets used in the *continual name-only* classification scenario is maximally $5.5\times$ larger than the manually curated ones, whereas in the previous scenario it was $144\times$ larger. This arises from the fact that the continual datasets we compare against are already big in size.

we collect through our webly-supervised approach are more recent, not necessarily centered, and of significantly higher image quality. To bridge this gap and align the two datasets, we employ downsampling techniques to resize our images to $32 \times 32$ pixels. This downsampling process proves to be effective in improving the performance of our webly-supervised approach on CIFAR100, specifically when using a normal budget. We observe a notable enhancement in performance, with an approximate 8% increase from 30.3% to 38.7%.

**Scale of Datasets.** Examining the statistics of continual datasets presented in Table 16, we observe an interesting contrast. Unlike the datasets collected in the traditional *name-only* classification setup, the scale of the datasets used in the *continual name-only* classification scenario is maximally $5.5\times$ larger than the manually curated ones. This stands in a big contrast to the previous scenario (*name-*

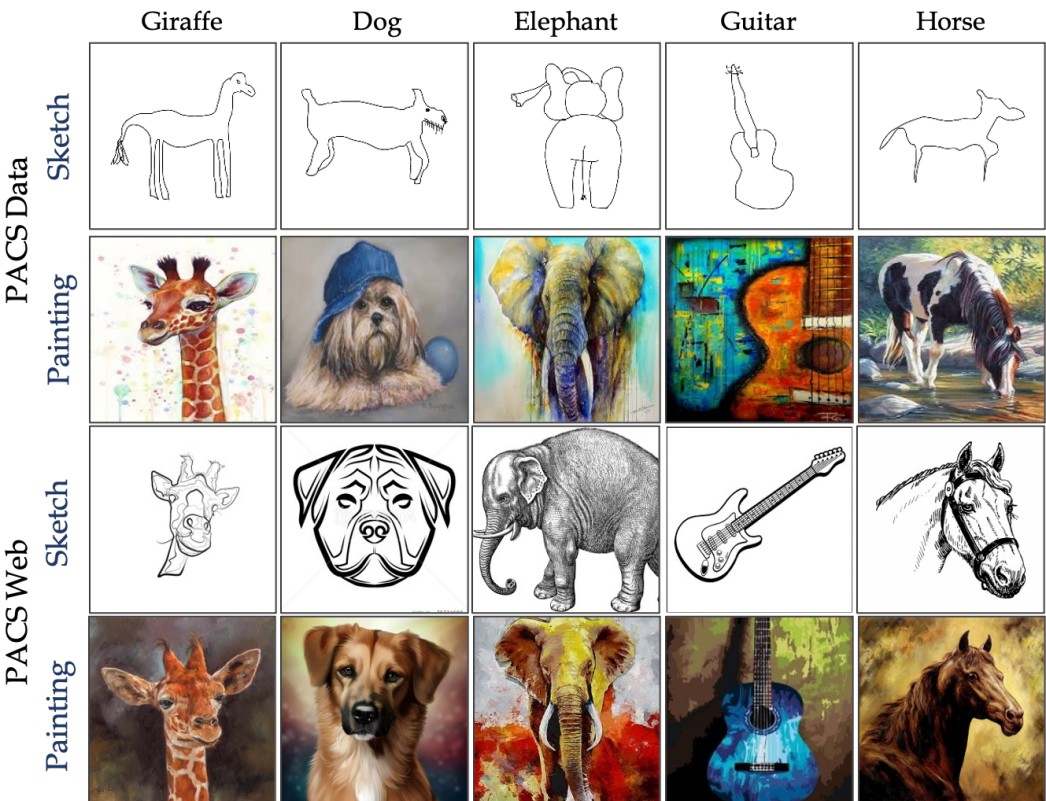

Figure 5: **Comparison of PACS and Webly-Supervised Data.** The upper two rows depict the sketch and painting domains of the manually annotated PACS dataset, while the last two rows showcase the sketch and painting domains of our webly-supervised data. Although there is some resemblance between the painting domains of both datasets, a significant domain gap becomes apparent when comparing the sketches. In the PACS dataset, sketches refer to quick drawings, whereas in our web search, sketches are composed of line drawings and detailed sketches.

*only* classification), where the scale was $144\times$ larger. The reason behind this disparity lies in the fact that the continual datasets we compare against are already substantial in size.

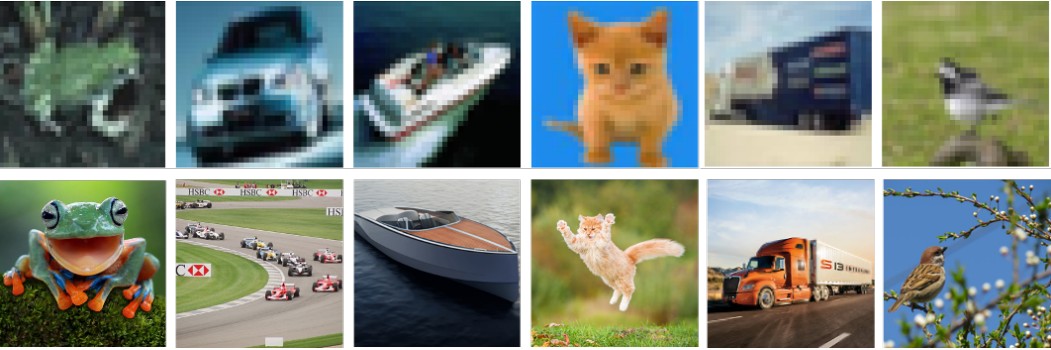

Figure 6: **Comparison of CIFAR100 and Webly-Supervised Data.** The upper row displays $32 \times 32$ samples obtained from CIFAR100, while the second row showcases images collected through our webly-supervised approach. Evidently, a substantial domain gap exists between the CIFAR100 images, characterized by their low quality, centered composition, and outdated nature, and the samples we collect, which are recent, not necessarily centered, and of significantly higher image quality. To bridge this gap, we employ downsampling techniques to resize our images to $32 \times 32$. It is noteworthy that through this downsampling process, we achieve an improvement in the performance of our webly-supervised approach on CIFAR100 with a normal budget, boosting performance by approximately 8% from 30.3% to 38.7%.

## E  ETHICS

We took steps to ensure that we do not violate copyright laws and avoiding explicit content from the internet. Steps for avoiding explicit content from the internet are detailed in Section 3 in "How do we prevent unintentional download of explicit images?". We clarify that we will only distribute links of the images for all C2C datasets with the class-names, including EvoTrends for reproducibility. Distributing links of images does not violate copyright laws while allowing reproducibility of our method. Similarly, to the best of our knowledge, training classification models on copyrighted data for this work is covered by copyright laws in UK [6].

---

[6]https://www.gov.uk/guidance/exceptions-to-copyrightnon-commercial-research-and-private-study

