# OpenReview forum: "From Categories to Classifier: Name-Only Continual Learning by Exploring the Web"
_ICLR.cc/2024/Conference — ICLR 2024 Conference Withdrawn Submission_

### Official Review · Reviewer_7o8D · 2023-10-28

**Soundness:** 2 fair
**Presentation:** 3 good
**Contribution:** 1 poor
**Rating:** 3
**Confidence:** 4

**Summary:**

This paper explores continual learning, where new categories can be introduced over time, in a setting where we have the names of the introduced categories but no curated labeled data for those categories. Instead the labels come from the web, ie a very large uncurated set of data that can be queried by category name, but where the precision and recall of the queries are not necessarily guaranteed. They show that their approach is only slightly less accurate than approaches trained on much smaller curated datasets. They also contribute a class-incremental dataset, which they call EvoTrends, which they were able to quickly curate from the web. The research contribution of the work is not very clear or strong, the main contributions are the engineering-heavy and expensive application of web supervised data to continual learning and an experimental analysis. There are significant ethical concerns with the use of web-scraped data for training that are not addressed by the authors.

**Strengths:**

The claims in the paper are clear and they show many experimental results and ablate different dimensions of the problem. The discussion of the performance gap and web-supervised limitations is a nice addition. This paper was clearly a significant implementation and engineering effort and is the first to my knowledge to run this scale of experimental analysis of web supervision for continual learning when provided with only the category names for each step.

**Weaknesses:**

The research contribution of the paper is not very clear compared to prior work showing that weakly labeled data from the web can be useful. The only difference here is that the exact same approach (using web queries to collect training data) is being done in a continual learning scenario. Comparison between models trained on very large webly-supervised data (which in many cases may contain the exact images and labels in the curated datasets, for example CUB was directly curated from flickr) as compared to smaller manually labeled datasets is not exactly a fair comparison, as the experts that provided the information are still very much a component of the first instance, just in this case they are given less credit for how that human expertise is a necessary component of the system to get to training data that can produce a useful signal. The increase in number of training images should be included for every experiment in Table 1 to provide context, and the overlap between the webly-supervised training and the curated training should be explicitly analyzed.

**Questions:**

It is not surprising that larger datasets with reasonably good but imperfect labels do ok, this has been well-demonstrated in the literature. What novel insights has this work given us into the use of very large weakly-labeled web data for classification? What about the continual learning setting the authors explore (where they query for classes, add the large weakly-labeled data to training, and train vanilla classification models or do nearest-neighbor classification per-timestep) is fundamentally different from just simple classifier training for a set of defined classes based on web-supervision without the continual setting?

Expert-in-the-loop curation of these larger datasets may only add a small amount of latency and significantly improve downstream performance, this is not discussed in the paper (see https://arxiv.org/abs/2302.12948)

**Details Of Ethics Concerns:**

Though Israel and Japan have passed laws making training models on web data regardless of license or data ownership legal, this is very much a complex and ongoing discussion and the legality in some countries does not make the ethical complexity moot. I would have appreciated a more nuanced handling of this complex issue from authors, as the rights to and ownership of data on the web and the use of that data without consent can be harmful, extractive, or even colonial, and is not legal or its legality is under discussion in many other places worldwide. It is unclear whether the proposed EvoTrends dataset complies with data copyright laws.

---

> ### Author Response · Authors · 2023-11-17
> **Rebuttal to Ethics Concerns**
>
> We address the ethics concern of copyright in training models and data release raised by the reviewer:
>
> **(a) Copyright**: We kindly request the reviewer to differentiate between motivation for future commercial applications based on recent trends in selected countries detailed in the introduction, with the research work itself. To the best of our knowledge (we are *not* legal experts), training *classification* models on copyrighted data for this research paper is perfectly legal practice and covered by copyright laws in UK [1] and EU [2] for academic research purposes, like most other papers which train models on vision datasets.
>
> **(b) Data Release**: We clarify that we will only distribute *links* of the images for all C2C datasets with the class-names, including EvoTrends for reproducibility. Distributing links of images is a perfectly legal practice and does not violate copyright laws.
>
> We hope this alleviates reviewer concerns. We added a Section E in Appendix highlighting these points.
>
> [1] https://www.gov.uk/guidance/exceptions-to-copyright#non-commercial-research-and-private-study
>
> [2] https://digital-strategy.ec.europa.eu/en/library/copyright-digital-single-market-brochure

---

> ### Author Response · Authors · 2023-11-17
> **Rebuttal**
>
> Thank you for your insightful review. We appreciate your recognition of our paper's clarity, thorough experimental analysis, and discussion on web-supervised limitations. Your acknowledgment of the significant effort in our engineering and implementation is highly encouraging. We will address the concerns raised in the following responses:
>
> **W1 and Q1 [Research Contribution & Novelty]**. We respectfully disagree with the reviewer and would like to push back strongly on this claim. We claim that:
>  - *[Webly supervised learning for learning targeted tasks.]* We adapt webly-supervised data to be used in downstream tasks. To the best of our knowledge, webly supervised learning has been primarily used as a pretraining task in past literature. Use in downstream tasks often entails targeted collection of data instead of a broad sweep across the internet. See recent work "Internet Explorer: Targeted Representation Learning on the Open Web" for a detailed outlook.
> - *[Impactful Solution.]* Previous works often introduce algorithmic novelty by designing ad-hoc data collection components using large vision-language models; we show that revisiting webly supervised learning is far better than current hot trends of using retrieval based approaches and synthetic data generation in a name-only classification setting.
> - *[Extensive Engineering.]* We alleviate past shortcomings in webly supervised learning by introducing new safeguards, optimize the scraping pipeline, re-examine various components for webly-supervised learning to name-only settings such as data sampling, effectiveness of various search engines, and that curation of webly-supervised data is unnecessary as linear probes are insensitive to this noise.
> - *[Finding shortcomings in continual learning and alleviating them]*. We uncover important shortcomings in continual literature along with extensively engineering a simple method that can have a major real-world impact.
> - *[Analysis of why our method works, and what is still missing]*: We provide not only strong results, but also isolate the reasons for improvement from our method by presenting an analysis on the impact of scale, effect of noisy samples, effect of model architectures, etc in our work with counterintuitive findings. We provide a detailed qualitative and quantitative overview of distribution shifts in the continual datasets in Appendix D to highlight the key bottlenecks in continual name-only learning.
>
> **[Humans involved in webly-supervised learning]** We do not understand the question. We acknowledge that internet is the creation of human labour and algorithmic systems. We believe it is quite likely that human experts used Flickr to initially filter samples and not vice-versa. We are not aware of any mechanisms to index these datasets back explicitly. Computing a match fraction is flawed in the sense it does not isolate the samples indexed back, rather than initially filtered which is perfectly valid.
>
> **[Expert-in-the-loop].** Thanks for bringing this to our notice. We have cited it along with other active learning approaches as they are  relevant works! However we note the following:
> - SuS-X and Neural priming have a similar nearest neighbor based pipeline (but no human labels in loop) with comparable or higher gains to a zero-shot CLIP model to our best knowledge. Exact comparisons between these methods was unfortunately not possible as the authors of aforementioned expert-in-the-loop work could not share their collected data.
> - We, in turn, outperform both SuS-X and Neural priming consistently by very wide margins across many downstream datasets and across architectures.

---

### Official Review · Reviewer_M9S2 · 2023-10-30

**Soundness:** 2 fair
**Presentation:** 2 fair
**Contribution:** 1 poor
**Rating:** 5
**Confidence:** 5

**Summary:**

This paper proposes a novel paradigm termed name-only continual learning to overcome continuous learning's reliance on time-consuming and expensive large-scale annotated datasets. Specifically, this paper proposes to leverage the expansive and ever-evolving internet to query and download uncurated webly-supervised data for image classification. It shows the potential of using uncurated webly-supervised data to mitigate the challenges associated with manual data labeling in continual learning. The experimental results show that it investigates the reliability of the web data and finds them comparable, and in some cases superior, to manually annotated datasets. Besides, it consistently exhibits a small performance gap in comparison to models trained on manually annotated datasets when applied across varied continual learning contexts.

**Strengths:**

1. The paper is well-written and has a clear structure.
2. This paper provides a new method to boost the performance of image classification. It shows that using uncurated webly-supervised data can significantly reduce the time and expense associated with manual annotation in the proposed name-only continual learning setting.

**Weaknesses:**

1. This paper is not innovative enough. It mainly explores a viable method of utilizing the practical applications of web data, and the innovative issues involved are insufficient.
2. It is not convincing that the paper attributes the performance improvement to the dataset sizes of the uncurated, noisy, and out-of-distribution web data. The paper needs to provide more experimental evidence about the impact of these three issues of web data, especially the large number of noisy samples that will definitely lead to a decrease in model performance.

**Questions:**

see weaknesses

---

> ### Author Response · Authors · 2023-11-16
> **Rebuttal**
>
> Thank you for your valuable feedback. We are grateful for your recognition of our paper's structure, and real-world impact of reducing the time and expense associated with manual annotation and for acknowledging our novel paradigm of name-only continual learning. We address the concerns raised below:
>
> **W1. [Not Innovative]**: We respectfully disagree with the reviewer and would like to push back strongly on this claim. As we detailed in our Introduction section, and highlighted by Reviewer XoLU, 7ydL we highlight that:
> 1) Study a Pressing Problem: Continual learning methods traditionally ignore training data creation which creates a huge cost bottleneck. We bring this to notice of the community, formalize this problem and propose a way to alleviate this bottleneck. Our solution is applicable across an extensive range of continual learning scenarios such as class, domain and time-incremental.
> 2) Impactful Solution: Even in the static name-only classification scenario where we can compare fairly with alternative solutions proposed in literature, we move the field forward by very large margins of performance (10-25%) where other approaches listed in Table 3 and Appendix C published in CVPR’23, ICCV’23 and NeurIPS’23 improve upon previous works by 1-3% margins.
> 3) Analysis of why our method works, and what is still missing: We provide not only strong results, but also isolate the reasons for improvement from our method by presenting an analysis on the impact of scale, effect of noisy samples, effect of model architectures, etc in our work with counterintuitive findings. We provide a detailed qualitative and quantitative overview of distribution shifts in the continual datasets in Appendix D to highlight the key bottlenecks in continual name-only learning.
>
> We request the reviewer to reconsider and not dismiss our hard work with a claim of “not innovative enough”.
>
> **W2. [Impact of noisy samples not studied]**: We acknowledge the reviewer's perspective and clarify that our work already provides an extensive analysis on factors like: Scale in Table 2, Noisy samples across different dimensions like duplicity and irrelevant samples in Appendix B.1 (as requested by the reviewer), different architectures in Appendix B.1, different search engines in Appendix B.1, class-balancing as web samples are highly imbalanced in Appendix B.1. We hope this analysis alleviates the concern of the reviewer.
>
> We would be happy to answer any further questions you have. If you do not have any further questions, we hope that you might consider raising your score.

---

### Official Review · Reviewer_7ydL · 2023-10-30

**Soundness:** 2 fair
**Presentation:** 3 good
**Contribution:** 3 good
**Rating:** 6
**Confidence:** 5

**Summary:**

This paper proposes name-only continual learning, where category names are used to query and download uncurated webly-supervised data for continual learning. The main finding is that models trained on uncurated webly-supervised data can benefit from the large data scale, and the performance can equal or even surpass the that of models trained on manually annotated datasets. Performance is also shown to be better than that of state-of-the-art name-only classification approaches. In different continual learning scenarios where only class names are available, the observations remain similar. Finally, a new continual learning dataset EvoTrends is introduced to include trending products year-by-year from 2000 to 2020.

**Strengths:**

- This is generally a solid paper that presents the first name-only continual learning approach using uncurated webly-supervised data. I like the idea a lot, the method proves to be working well, and the analysis/comparisons are comprehensive.
- The continual learning dataset EvoTrends is a good plus, offering a more natural class-incremental scenario to train and evaluate continual learners.

**Weaknesses:**

- One main concern about this paper is the missing detail/analysis of the continual learning algorithm itself (see questions below).
- Another concern is that many analysis results under the non-continual name-only classification setting (Section 4) won't necessarily translate to the continual learning setting. For example, Section 4 provides nice ablations on the model architecture and pretraining algorithm (Table 8), ensembling from multiple web engines, data cleaning and class balancing strategies - will they make a larger impact on continual learning? Also, under continual learning setting, it's unclear if the performance will remain competitive with those name-only classification techniques.

**Questions:**

- My understanding about the continual learning algorithm used in this paper is fixed feature extractor + retraining classifier every timestep t. But how is the retraining actually implemented? At the end of Section 3, it mentions "the classifier is trained not only on the uncurated data downloaded from the current timestep t but also on the uncurated data downloaded from all prior timesteps" which would be pretty costly. In Section 5.1, it mentions "we implement experience replay and utilize class-balanced sampling to select training batches from the previously collected samples". Does this mean the authors just use the traditional continual learning method based on experience replay (just with some smart sampling)? No more whistles and bells?
- In either case, how well is the adopted algorithm with web data addressing the catastrophic forgetting in the class-incremental setting? What about the domain-/time-incremental settings where there can be overlapping classes with domain shifts? How is the adopted algorithm handling those cases?
- Last question about Table 1: finetuning the backbone 1) achieves much better end performance than not, for both MA data and C2C. 2) The benefits of C2C over MA data are reduced a lot (except for FGVC). Is that also the case for continual learning? If true, that makes C2C less interesting when finding the best end performance is the goal.

---

> ### Author Response · Authors · 2023-11-17
> **Rebuttal**
>
> Thank you for your valuable feedback. We are grateful for your recognition of our paper's quality and for acknowledging our as the first work to introduce the continual name-only setting. Your appreciation of the in-depth analysis and comparisons in our work is highly encouraging. Additionally, we are happy that you liked the merit of EvoTrends as a more realistic approach to class-incremental continual learning.
>
> We address your concerns below:
>
> **W1 [On Continual Learning Algorithm]**
>
> *Q1. [Retraining]* We provide the following details to give more clarity: At each timestep t, the classifier (a single linear layer) is simply initialized based on the linear model at task t-1, and updated quickly with experience replay on past samples with random sampling. There are no extra bells and whistles.
>
> Why do we use this? This was shown to be state-of-the-art in scenarios with limited compute but unrestricted memory (Prabhu et al, 2023a). We follow the same setup, fixing the number of FLOPs and hence computational cost into tight, normal and relaxed regimes.
>
> We agree this could be expensive, hence additionally present the NCM and approximate kNN baselines which take the classifier at task t-1 and simply add the new samples to it. Hence, they can operate on “tight” computational budgets being some of the cheapest approaches to implement.
>
> However, due to the linear classifier being updated only on extracted features and no prior samples are stored this process is relatively fast in practice. Compared to finetuning the whole backbone, training a linear classifier is extremely efficient, emphasized  below.
>
> *Q2. [Evaluation of Domain/Time-Incremental Setting]* We were not sure if we understood the question correctly.
> - How do we query the web? As detailed in Section 5.1, for domain-incremental setting, we query the web with class-names + domain-names, but consider them in the same class according to class-names. For time-incremental settings, we query the web using timestamp queries to limit time ranges and only consider class-names for the labels.
> - How is catastrophic forgetting alleviated? RanPAC [1] and ACM (Prabhu et al., 2023b) already show the effectiveness of NCM and ACM respectively for specific domain and time incremental scenarios. Inspired by this, we benchmark those methods in a name-only continual learning setting and find that kNN outperforms linear classification which outperforms NCM in a name-only setting. The effectiveness of simple approaches is quite underrated in tackling catastrophic forgetting is quite underrated in the continual learning domain.
>
> [1] RanPAC: Random Projections and Pre-trained Models for Continual Learning
>
> *Q3. [Finetuning]* This is a great question! While finetuning achieves better performance, we clarify that it is dramatically more expensive. We could not satisfactorily train it on our continual learning setup, even with 10x computational resources compared to the  relaxed compute budget. Hence, its usefulness is very limited. Hope this addresses the concern of the reviewer.
>
> We hope we were able to clarify all your concerns in W1.
>
> **[W2] Analysis results would not translate to a continual learning setting.**
> - We are not sure why the ablations on noise, multiple web engines, data cleaning and class balancing etc would not translate as they are quite dataset independent. We did a detailed analysis of the distribution shift to detail the precise domain gaps between web datasets and annotated training data in Appendix D.
>
> *What did we aim for in a continual learning setup?*
> - We had a high bar for compute efficiency so we focused on those aspects. Note that latest approaches which use stable diffusion models to generate images or retrieve relevant images from a 5B scale dataset by text are far more expensive than the budgets allowed in our relaxed setting.
>
> We would be happy to answer any further questions you have. If you do not have any further questions, we hope that you might consider raising your score.

---

### Official Review · Reviewer_xW5K · 2023-11-01

**Soundness:** 3 good
**Presentation:** 4 excellent
**Contribution:** 2 fair
**Rating:** 5
**Confidence:** 2

**Summary:**

This study introduces "name-only continual learning," a new approach in Continual Learning (CL) that overcomes the limitations of time and cost in creating extensive annotated datasets by utilizing uncurated webly-supervised data for image classification. The proposed method not only achieves comparable, sometimes even superior, results to manually annotated data but also delivers a significant performance boost in accuracy, along with the development of EvoTrends.

**Strengths:**

+ A valuable exploration of using uncurated webly-supervised data for fine-grained classification and incremental learning.

+ A well-written and structured paper with insights on the usage of uncurated webly data on visual learning tasks, and its advantage over other CLIP-based methods.

**Weaknesses:**

- The experiments fail to reflect popular incremental setup, including restricted access to the data from previous tasks.

- The technical sections are rather weak, and limited details about CL approaches were exposed. It's unclear if the application on CL is reasonable.

- The advantage of the "uncurated webly-supervised" may be limited to a few fine-grained classifications where only small numbers of training data are available.

**Questions:**

1.	One of the major concerns is the setup for CL experiments. CL is a well-established field, but very few competitive methods are reported in the paper. Could authors add a few more recent CL methods for comparison?
2.	CL usually has strong restrictions on access to the “old” data from previous tasks. Nonetheless, this work uses data from all tasks to “re-train” each time. Did I miss any details?
3.	The comparison with other “name-only” and CLIP methods was not well-explained. The advantage of the proposed method, as claimed by the authors, is the “scale.” In fine-grained classification, a sufficient number of samples/class are critical. I'm not sure if any off-the-shelf CLIP model is supposed to do this well without any special treatment.
4.	Regarding regular CL tasks on regular visual concepts or objects, the accuracy is only comparable to, or worse than, MA data. It means not only the quantity but also the quality of the images matter in these tasks. While the paper attempts to highlight name-only CL, it seems this approach did not work well in CL. BTW, CL usually limits the number of samples per class, too.

---

> ### Author Response · Authors · 2023-11-16
> **Rebuttal**
>
> We thank the reviewer for appreciating our exploring of webly-supervised data for fine-grained classification and continual learning, the quality of our work, and our provided insights on the power of webly-supervised data for visual learning tasks.
>
> We address your concerns below:
>
> **W2. [On Restricted Memory].** This is a great question. We clarify here that while most prior continual learning approaches have the assumption that data cannot be fully stored either due to privacy concerns, when this happens is underspecified (a memory buffer is used to store a limited number of samples of choice). Our framework allows us to study the validity of this data constraint – Since we retain only links of images, we can access past data whenever we can by redownload the links. If the content is removed from the link, we can no longer access it. We could redownload 100% of the data within the span of a few weeks when we ran the final experiments. Longer timescale works of (Cai et.al., 2021) find that they could re-download 95+% of the links last collected in 2014 entailing that links typically do not get removed.
>
> Hence, we rely on the setup constraints introduced in prior work which argues against restricted memory (Prabhu et. al, 2023a) claiming a wide range of real-world constraints can have relaxed memory restrictions. We clarified this in our paper draft in Section 5.1 Training models.
>
> **W1 [Added comparisons with CL Approaches].** We clarify here that previous work (Prabhu et.al., 2023a) showed that in computationally budgeted setup with relaxed memory constraints, traditional continual learning methods fail to be effective and simple experience replay baseline performs the best. Hence, we did not study the past methods, but additionally included baselines such as kNN and NCM which have been shown to work well in similar settings (Prabhu et. al., 2023b). However, we highlight that we do not propose any new continual learning methods but focus on ways of collecting training data, extensive comparisons with CL methods is somewhat orthogonal to the focus of our work.
>
> **W3 [Comparison with zero-shot CLIP approaches unfair].** We agree with the reviewer. We additionally compare with methods which also collect data from large datasets like LAION5B leveraging CLIP models in Table 3, alongside the data-free methods. We compared data-free methods as it is a valid method of performing name-only classification, but as reviewer notes and we present in Table 3, methods which collect training data outperform methods which do not use training data. We note that we outperform the latest methods which also collect data along with large models (presented in CVPR ‘23, ICCV’23 and NeurIPS ‘23) by large margins.
>
> **W4 [Continual Learning Results worse than MA, Quality Reasons].** This is a great question. We clarify here that the performance on manually annotated datasets serves as an upper bound and not a fair comparison as they require a very expensive curation process, the collection has biases making them well-aligned with the test distribution as both sets are created after labeling by subsampling data. Reproductions of the test set creation of CIFAR10 and ImageNet [1] show that just by recreating the test set following the same procedure as originally followed, they observe 5-10% drops in performance of models despite no training data drifts etc. We updated the draft to highlight this point.
>
> In Appendix D, we investigate the performance drop between our webly-supervised data and annotated data in detail and find distribution gaps between the manually annotated test set and our training set. For example, for CLEAR time-incremental dataset we show that the buses there are interior of buses whereas when we scrape samples we scrape exterior of buses. Similarly, for PACS, manually annotated sketches are simple quick doodles compared to webly-supervised sketches which are pencil sketches. This indeed creates an extra gap between our training data and the test data.
>
> [1] Recht et. al, Do ImageNet Classifiers Generalize to ImageNet?
>
> We would be happy to answer any further questions you have. If you do not have any further questions, we hope that you might consider raising your score.

---

### Official Review · Reviewer_xoLu · 2023-11-01

**Soundness:** 3 good
**Presentation:** 4 excellent
**Contribution:** 3 good
**Rating:** 6
**Confidence:** 4

**Summary:**

This paper explores the name-only continual learning, which combines the webly-supervised learning with continual learning using a pre-trained model. In this paper, it is found that training with webly collected data can be comparable or even superior to training with manually annotated datasets. Besides, a new class-incremental dataset to capture real-world trends is introduced.

**Strengths:**

1. The new setting is meaningful for reducing the cost of manual annotation of new datasets.
2. This paper presents outstanding results with webly-supervised learning under single-task and continual learning.
2. This paper extends the webly-supervised learning to continual setting, which is reasonable for real-world application. A new benchmark with ever-changing real-world data is introduced for webly-supervised / name-only continual learning.
3. Although somewhat counterintuitive, it is interesting to see that a pre-trained model can learn so well with noisy, uncurated data. This paper finds data scale to be the key.
4. The evaluations in this work are thorough. Multiple fine-grained vision datasets are included. Cost-aware evaluation for CL is used, and the results are significant.

**Weaknesses:**

1. In the last paragraph of Section 3, it is said that for the continual name-only setting, "Once we complete downloading the data, the classifier is trained not only on the uncurated data downloaded from the current timestep $t$ but also on the uncurated data downloaded from **all prior timesteps**." If data from all timesteps are used, it would not strictly conform to the definition of continual learning. Section 5 also mentions a class-balanced sampling to select training batches from the previously collected samples. Accessing previous samples would violate the definition of CL. If just a few data are used, to what scale is the method selected?
2. In section 3, this paper mentions *we implemented some cost-effective safeguards*. However, no details are found in this paper (only one following sentence that refers to the safe-search feature of search engines).
3. (Minor) The results on other network architectures (for example, ViT) are worse then ResNet, especially for the Cars dataset.
4. (Minor) In section 5.2, when explaining the performance gap, it is said that "The current training operates within a tight computational budget". Here, the word "tight" is ambiguous since the results it mentions are under the "normal" instead of the "tight" budget suggested in Table 5.

**Questions:**

See questions in the Weaknesses part.

**Details Of Ethics Concerns:**

This paper includes large-scale data collection and possible data release with limited restrictions. Although it mentions a "cost-effective safeguard" using the "safe-search feature of search engines", it still may (with a high possibility) collect unintentional data from the internet.

---

> ### Author Response · Authors · 2023-11-16
> **Rebuttal**
>
> We thank you for the constructive feedback and the positive aspects you highlighted in our work. We are pleased that you found our approach in reducing manual annotation costs, the outstanding results in webly-supervised learning under both single-task and continual learning, and the application of our methods to real-world scenarios useful. Your recognition of our counterintuitive findings on learning with noisy data and the thoroughness of our evaluations, especially in the context of multiple fine-grained vision datasets and cost-aware continual learning, is motivating.
>
> We address your concerns below:
>
> **W1. [Access to Past Data].** This is a great question. We clarify here that while most prior continual learning approaches have the assumption that data cannot be fully stored either due to privacy concerns, when this happens is underspecified (a memory buffer is used to store a limited number of samples of choice). Our framework allows us to study the validity of this data constraint – Since we retain only links of images, we can access past data whenever we can by redownload the links. If the content is removed from the link, we can no longer access it. We could redownload nearly 100% of the data within the span of a few weeks when we ran the final experiments. Longer timescale works of (Cai et.al., 2021) find that they could re-download 95%+ of the links last collected in 2014, entailing that linked images typically do not get removed.
>
> Hence, we rely on the setup constraints introduced in prior work which argues against restricted memory (Prabhu et. al, 2023a) claiming a wide range of real-world constraints can have relaxed memory restrictions. We clarified this to our paper draft in Section 5.1 Training models.
>
> **W2. [Cost Effective Safeguards].** We apologize for not detailing this in more detail, but emphasize that we had seriously considered  this ethical concern due to past mishaps (Birhane and Prabhu, 2021) and tried our best to alleviate it.  In our paper we use two particular safeguards: (1) Using safe-search filtering (as mentioned) and (2) Making sure that the category-name is written to ensure no ambiguity in the search query that could be understood as a non-safe content. For example, we would correct "african ass" to "african wild donkey" or "Equus africanus" to avoid inadvertently collecting explicit content. We note that adding the auxiliary suffix as stated in "How to design queries the web?" also greatly helped filter unwanted contexts cheaply. We manually inspected a few hundred of the downloaded images for data quality with random sampling and found that strict safe-search filtering in web search engines by itself is extremely effective. We updated the draft with the second part in Section 3 Step 1 and added an explicit Ethics subsection in Appendix E.
>
> **W3. [ResNet vs ViT].** In Table 8 (Appendix B), we show the impact of various network architectures and pretraining algorithms. ViT based models did always outperform the ResNet models, including on the Cars dataset– we are not sure if we understood the the reviewer's concern correctly.
>
> **W4. [Typo].** Thanks for bringing it to our notice, we fixed it in the updated draft.
>
> We would be happy to answer any further questions you have. If you do not have any further questions, we hope that you might consider raising your score.